# DNT: A Deeply Normalized Transformer that can be trained by Momentum SGD

**Xianbiao Qi**[1], **Marco Chen**[2], **Wenjie Xiao**[3], **Jiaquan Ye**[1], **Yelin He**[1], **Chun-Guang Li**[*4],
**Zhouchen Lin**[*5,6]

[1]Intellifusion Inc.    [2]Tsinghua University    [3]Johns Hopkins University

[4]Beijing University of Posts and Telecommunications

[5]State Key Lab of General AI, School of Intelligence Science and Technology, Peking University

[6]Institute for Artificial Intelligence, Peking University

## Abstract

Transformers have become the de facto backbone of modern deep learning, yet their training typically demands an advanced optimizer with adaptive learning rate like AdamW, rather than a momentum SGDW (mSGDW). Previous works show that it is mainly due to a heavy-tailed distribution of the gradients. In this paper, we introduce a Deeply Normalized Transformer (DNT), that is meticulously engineered to overcome the heavy-tailed gradients issue, enabling seamless training with vanilla mSGDW while yielding comparable performance to the Transformers trained via AdamW. Specifically, in DNT, we strategically integrate normalization techniques at proper positions in the Transformers to effectively modulate the Jacobian matrices of each layer, balance the influence of weights, activations, and their interactions, and thus enable the distributions of gradients concentrated. We provide both theoretical justifications of the normalization technique used in our DNT and extensive empirical evaluation on two popular Transformer architectures (*i.e.*, ViT and GPT), validating that: a) DNT can be effectively trained with a vanilla mSGDW; and b) DNT outperforms its counterparts.

## 1 Introduction

Transformer (Vaswani et al., 2017) has revolutionized numerous domains in artificial intelligence, demonstrated remarkable capabilities across natural language processing (Radford et al., 2018; 2019; Brown et al., 2020; Dubey et al., 2024; Team, 2023; Liu et al., 2024), computer vision (Dosovitskiy et al., 2020; Liu et al., 2022; Dehghani et al., 2023), AIGC (Ramesh et al., 2021; Peebles & Xie, 2023), and multi-modal applications (Li et al., 2022; Liu et al., 2023a) and become the de facto backbone of modern deep learning.

Nowadays, it is widely accepted that Adam (Kingma & Ba, 2014) or its descendant AdamW (Loshchilov & Hutter, 2019) are the standard optimizer for training Transformers; whereas the classical SGD (Robbins & Monro, 1951) and its variants (Nesterov, 1983; 1998; Johnson & Zhang, 2013), *e.g.*, momentum SGD (mSGD), usually under-perform when training Transformers. Despite of its heavier load on GPU memory than mSGD, Adam is used as the optimizer in most recent studies on Large Language Models (LLMs) (Dubey et al., 2024; Team, 2023; Liu et al., 2024) and multi-modal models (Li et al., 2022; Liu et al., 2023a). Naturally, an interesting question arises:

> *Can Transformers be trained via mSGD to yield performance matched to that is trained via Adam? Or, under what conditions?*

To answer these questions, we need to understand why mSGD typically underperforms Adam when training Transformer. Previous studies (Simsekli et al., 2019; Zhang et al., 2020) reveal that the fundamental reason lies in the statistical property of the stochastic gradients in Transformer

---

∗ Corresponding authors

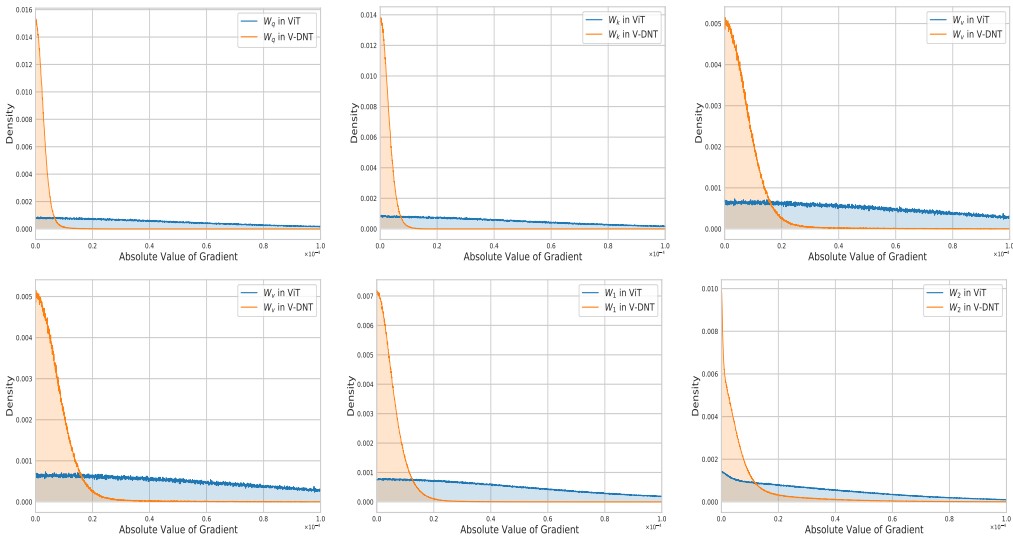

FIGURE 1: Distributions of the absolute values of the entries in gradients for ViT with PreNorm (marked in blue) and our V-DNT (marked in orange), where V-DNT denotes the vision variant of our DNT. We observe that the amplitude of the gradients in our V-DNT are typically quite small and well concentrated; whereas the gradient distributions of the standard ViT have a long tail.

architectures. Unlike Convolutional Neural Networks (CNNs) (LeCun et al., 1998; He et al., 2016) that are trained on tasks like ImageNet, where the entries of the gradients are typically small and well-concentrated, the gradients of Transformer typically exhibit heavy-tailed distributions, as shown in blue in Figure 1. This heavy-tailed distribution means that the amplitudes of the gradient entries span a wide range and thus it is hard to keep step with each others when updating weights. Thus, Adam uses a normalized term between the first-order term (i.e. gradients) and the square-root of the second-order term. Owing to the normalization, Adam is robust to the heavy-tail distribution of the gradients. This explains why Adam has become the standard optimizer for training Transformer in practice. On contrary, mSGD directly uses the first-order gradient with momentum to update the weights, and thus the weights updating suffers from the difficulty to keep pace with each others. Consequently, an interesting question turns out to be: can we help mSGD to relieve the issue of heavy-tail gradients in training Transformers? And how?

To mitigate the issue of heavy-tail gradients in training Transformers with mSGD, we propose to add or adjust the positions of normalization operations in Transformers, motivated by analyzing the Jacobian matrix of different modules. Roughly speaking, we proposed to use the properly positioned normalization operator to amend the Jacobian matrix of $\frac{\partial \boldsymbol{y}}{\partial \boldsymbol{x}}$ less affected by the weights, the activations, or the joint influence of both weights and activations.

As illustrated in orange in Figure 1, we observe that our designed architecture, a Deeply Normalized Transformer (termed as DNT), exhibits a more concentrated gradient distribution than its counterpart which has a heavy-tailed distribution. In this paper, we provide not only theoretical justification for the properly positioned normalization operator in our DNT, but also empirical evaluations to further validate that our DNT outperforms its counterparts, *i.e.*, ViT and GPT, on ImageNet classification and OpenWebText tasks. Since that the distributions of the gradients of DNT are more concentrated, training it with the vanilla mSGD can yield performance on par with that by an Adam optimizer.

To the best of our knowledge, this is the first work to show that using a vanilla mSGD can train a Transformer to achieve performance comparable to that of using Adam—provided that the Transformer architecture is properly modified to mitigate the issue of heavy-tail gradients.

## 2 PRELIMINARIES

This section will provide some preliminaries on high-dimensional random vectors, which enjoy many nice properties that are different from their low-dimensional counterparts. Two simple yet useful theorems are introduced below. Proofs can be found in Lemma 3.2.4 of (Vershynin, 2018).

**Theorem 1** (Concentration of norm). *Let $\boldsymbol{x}$ be an isotropic random vector in $\mathbb{R}^d$. Then, we have $\mathbb{E}\|\boldsymbol{x}\|_2^2 = d$. Moreover, if $\boldsymbol{x}$ and $\boldsymbol{y}$ are two independent isotropic random vectors, then $\mathbb{E}\langle \boldsymbol{x}, \boldsymbol{y}\rangle^2 = d$.*

**Theorem 2** (Almost orthogonality of high-dimensional independent vectors). *Let us normalize the random vectors $\boldsymbol{x}$ and $\boldsymbol{y}$ in Theorem 1, setting $\overline{\boldsymbol{x}} := \frac{\boldsymbol{x}}{\|\boldsymbol{x}\|_2}$ and $\overline{\boldsymbol{y}} := \frac{\boldsymbol{y}}{\|\boldsymbol{y}\|_2}$, in a high-dimensional space, the independent and isotropic random vectors $\overline{\boldsymbol{x}}$ and $\overline{\boldsymbol{y}}$, tend to be almost orthogonal,*

Theorem 1 establishes that $\|\boldsymbol{x}\|_2 \asymp \sqrt{d}$, $\|\boldsymbol{y}\|_2 \asymp \sqrt{d}$ and $|\langle \boldsymbol{x}, \boldsymbol{y}\rangle| \asymp \sqrt{d}$ with high probability, which implies that the cosine of the angle $\theta$ between two random vectors $\boldsymbol{x}$ and $\boldsymbol{y}$ satisfies $|\cos(\theta)| \asymp \frac{1}{\sqrt{d}}$. Theorem 2 implies that in high-dimensional space (*i.e.*, $d$ is very large), two random vectors are almost orthogonal. Thus, given $\boldsymbol{z} = \boldsymbol{x} + \boldsymbol{y}$ where $\boldsymbol{x}$ and $\boldsymbol{y}$ are two high-dimensional random vectors, we have $\|\boldsymbol{z}\|_2 \asymp \sqrt{\|\boldsymbol{x}\|_2^2 + \|\boldsymbol{y}\|_2^2}$.

**Jacobian of normalization.** Normalization (Ioffe & Szegedy, 2015; Ba et al., 2016; Zhang & Sennrich, 2019) is a widely used technique in deep learning. For example, LayerNorm (Ba et al., 2016) is defined as $\mathrm{LN}(\boldsymbol{x}) = \boldsymbol{\gamma} \odot \frac{\sqrt{d}\boldsymbol{y}}{\sqrt{\|\boldsymbol{y}\|_2^2+\epsilon}} + \boldsymbol{\beta}$, and $\boldsymbol{y} = \left(\boldsymbol{I} - \frac{1}{d}\boldsymbol{1}\boldsymbol{1}^\top\right)\boldsymbol{x}$, where $\epsilon > 0$ is a smoothing factor, $\boldsymbol{\gamma}$ and $\boldsymbol{\beta}$ are two learnable $\mathbb{R}^d$ vectors which are usually initialized to $\boldsymbol{1}$ and $\boldsymbol{0}$. Most recently, some recent LLMs (Touvron et al., 2023; Chowdhery et al., 2023; Team, 2023; Liu et al., 2024) uses RMSNorm (Zhang & Sennrich, 2019) to replace LayerNorm, where RMSNorm is defined as: $\mathrm{RMSN}(\boldsymbol{x}) = \boldsymbol{\gamma} \odot \frac{\sqrt{d}\boldsymbol{x}}{\sqrt{\|\boldsymbol{x}\|_2^2+\epsilon}}$, without the centering term and the bias term. The Jacobian matrix of RMSNorm with respect to $\boldsymbol{x}$ is calculated as follows:

$$\frac{\partial\,\mathrm{RMSN}(\boldsymbol{x})}{\partial \boldsymbol{x}} = \frac{\sqrt{d}}{\sqrt{\|\boldsymbol{x}\|_2^2 + \epsilon}}\,\mathrm{diag}(\boldsymbol{\gamma})\left(\boldsymbol{I} - \frac{\boldsymbol{x}\boldsymbol{x}^\top}{\|\boldsymbol{x}\|_2^2 + \epsilon}\right).$$

We use RMSNorm as our default normalization technique when mentioning of normalization, but our analysis can be generalized to the other normalization techniques. Here we use a numerator layout for all our gradients derivation throughout this paper.

**Stochastic Gradient Descent (SGD).** SGD (Robbins & Monro, 1951) is a classical and fundamental optimization algorithm in machine learning for training models by minimizing their cost functions. However, the vanilla SGD often suffers from slow convergence, especially in complex optimization landscapes with ravines, saddle points, or local minima. To address these limitations, momentum SGD (Nesterov, 1983; 2013; Sutskever et al., 2013) was introduced as an extension of the basic SGD algorithm. Momentum SGD (Nesterov, 1983; 2013; Sutskever et al., 2013) introduces a velocity term $\boldsymbol{m}$ that accumulates gradients over time, *i.e.*, $\boldsymbol{m}_{t+1} = \mu\boldsymbol{m}_t + \nabla L(\boldsymbol{w}_t)$, $\boldsymbol{w}_{t+1} = \boldsymbol{w}_t - \alpha_t\boldsymbol{m}_{t+1}$ where $\mu \in [0, 1)$ is the momentum coefficient that determines how much of the previous velocity is retained and $\alpha_t$ is the learning rate for the time step $t$. Unlike in the vanilla SGD, mSGD allows the optimization to build up a "momentum" in the direction of persistent gradient descent, which can effectively dampen the oscillations in high-curvature directions.

## 3 Theoretical Justification on Why DNT Can Be Trained with Momentum SGD

### 3.1 Problem 1: What is the root cause of heavy-tail distribution of gradients?

Previous works (Zhang et al., 2020; Simsekli et al., 2019) have pointed out that a heavy-tailed distribution of the stochastic gradients is a root cause of SGD's poor performance. Here, we will investigate this issue by analyzing the backpropagation of Transformers.

Suppose that $\boldsymbol{x}^{l+1} = f(\boldsymbol{x}^l)$ and we have obtained $\frac{\partial\mathcal{L}}{\partial \boldsymbol{x}^{l+1}}$ in a backpropagation process, then we calculate $\frac{\partial\mathcal{L}}{\partial \boldsymbol{x}^l}$ using a numerator layout as $\frac{\partial\mathcal{L}}{\partial \boldsymbol{x}^l} = \frac{\partial\mathcal{L}}{\partial \boldsymbol{x}^{l+1}}\frac{\partial \boldsymbol{x}^{l+1}}{\partial \boldsymbol{x}^l}$, where $\frac{\partial \boldsymbol{x}^{l+1}}{\partial \boldsymbol{x}^l}$ is called as the Jacobian matrix. Having had $\frac{\partial\mathcal{L}}{\partial \boldsymbol{x}^l}$, for any a forward layer with $\boldsymbol{x}^l = \boldsymbol{W}^l\boldsymbol{x}^{l-1}$, we compute $\frac{\partial\mathcal{L}}{\partial \boldsymbol{W}^l}$ as

$$\frac{\partial\mathcal{L}}{\partial \boldsymbol{W}^l} = \frac{\partial\mathcal{L}}{\partial \boldsymbol{x}^l}\boldsymbol{x}^{l-1\top} = \frac{\partial\mathcal{L}}{\partial \boldsymbol{x}^{l+1}}\frac{\partial \boldsymbol{x}^{l+1}}{\partial \boldsymbol{x}^l}\boldsymbol{x}^{l-1\top}. \tag{1}$$

---

Typical values for $\mu$ range from 0.9 to 0.99. In default, for all our experiments, we set $\mu$ to 0.90.

From Equation 1, we observe that the heavy-tail problem in gradients is indeed closely related to the Jacobian matrix $\frac{\partial \boldsymbol{x}^{l+1}}{\partial \boldsymbol{x}^l}$ when its singular values are very diverse. The Jacobian matrix suffers from highly diverse singular values for several reasons: 1) the weight matrix contains very diverse singular values; 2) the activations span widely, leading to Jacobians with very uneven singular value distributions. When a matrix has a wide range of singular values (*i.e.*, a very large condition number), it means that the transformation stretches the input very differently along different directions. Thus, during the backpropagation phase, the heavy-tail issue in the gradients occurs. *Therefore, a reasonable solution to relieve the heavy-tail issue is to constrain the uneven singular values of the Jacobian matrix by controlling the weight matrix and the activations.* This is the basic idea in this paper.

## 3.2 Problem 2: Mitigate the heavy-tail gradient problem by analyzing the Jacobian matrix

In this subsection, we will describe how we use different normalizations—adding or adjusting the position of the normalizations—to constrain the Jacobian matrix to relieve the heavy-tail gradient issue. *Note that we do not claim that we discover any new normalization methods, instead, we provide our understanding on how each normalization affects the Jacobian matrix.* We refer the readers to (Loshchilov et al., 2025; Zhu et al., 2025; Qi et al., 2025b; 2023) for more discussions about the normalizations. We will use red, green, blue, purple, magenta to associate with "InputNorm", "PreNorm", "MidNorm", "PostNorm" and "QKNorm", individually.

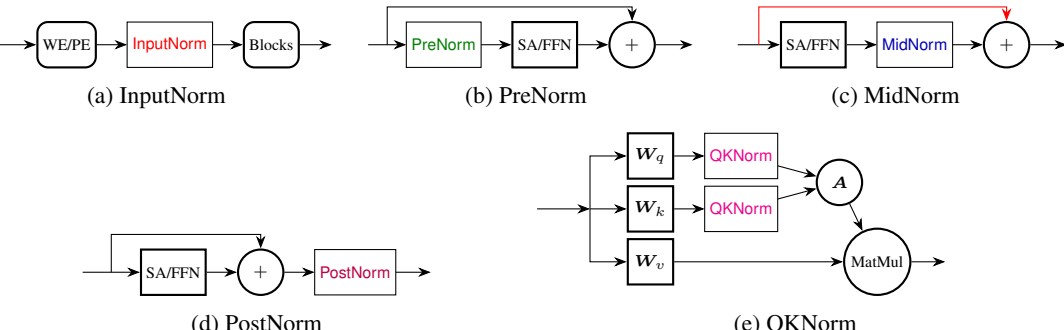

FIGURE 2: Five different normalization methods. The only difference between them is the position of normalization. In (A), "WE/PE" indicates Word Embedding (WE) and Patch Embedding (PE).

### 3.2.1 InputNorm

**Definition of InputNorm.** InputNorm in Transformer is defined as the normalization that is applied after the first word embedding in NLP or the first patch embedding in vision Transformer. As shown in Figure 2 (d), InputNorm is defined as

$$\boldsymbol{x}^0 = \text{InputNorm}(\boldsymbol{h}), \text{ where } \boldsymbol{h} = \text{Embedding}(\boldsymbol{i}), \tag{2}$$

where $\boldsymbol{i}$ is the input and $\text{Embedding}(\cdot)$ denotes the word embedding or the patch embedding. For a standard residual block in Transformer, we have that:

$$\boldsymbol{x}^{l+1} = \boldsymbol{x}^l + f(\boldsymbol{x}^l) = \boldsymbol{x}^{l-1} + f(\boldsymbol{x}^{l-1}) + f(\boldsymbol{x}^l) = \boldsymbol{x}^0 + f(\boldsymbol{x}^0) + f(\boldsymbol{x}^1) + \cdots + f(\boldsymbol{x}^{l-1}) + f(\boldsymbol{x}^l).$$

Each $\boldsymbol{x}^l$ will be the input into some modules, such as normalization, self-attention and feed-forward layers. The Jacobian matrices of some modules, such as LayerNorm and the dot-product self-attention, are sensitive to the norm of the input.

Under the assumption that random vectors are almost orthogonal in high dimension, we have that:

$$\|\boldsymbol{x}^{l+1}\|_2 \asymp \sqrt{\left( \|\boldsymbol{x}^0\|_2^2 + \|f(\boldsymbol{x}^0)\|_2^2 + \cdots + \|f(\boldsymbol{x}^l)\|_2^2 \right)}. \tag{3}$$

**Proposition 1** (Effect of the norm of the input embedding on the gradients)**.** *In a high-dimensional settings when all parameters and activations are high-dimensional, if the term $\|\boldsymbol{x}^0\|_2^2$ is very large, it will lead to gradient vanishing problem in all subsequent layers, provided that InputNorm is not used.*

It means that if the term $\|\boldsymbol{x}^0\|_2^2$ is large, then the norm $\|\boldsymbol{x}^{l+1}\|_2$ in each layer will also be large. If $\|\boldsymbol{x}^{l+1}\|_2$ is the input into a normalization layer, according to the Jacobian equation of normalization $\frac{\partial \mathrm{RMSN}(\boldsymbol{x}^{l+1})}{\partial \boldsymbol{x}^{l+1}} = \frac{\sqrt{d}}{\sqrt{\|\boldsymbol{x}^{l+1}\|_2^2+\epsilon}} \mathrm{diag}(\boldsymbol{\gamma}) \left( \boldsymbol{I} - \frac{\boldsymbol{x}^{l+1}\boldsymbol{x}^{l+1\top}}{\|\boldsymbol{x}^{l+1}\|_2^2+\epsilon} \right)$, the gradient flow in each layer will be significantly affected by the term $\|\boldsymbol{x}^0\|_2^2$. Thus, we need to constrain $\|\boldsymbol{x}^0\|_2^2$ before it is used as the input into the following layer.

**Remark 1.** *The $\|\boldsymbol{x}^0\|_2^2$ has a large influence of the gradient flow of the subsequent layers. If it is very large, it will lead to gradient vanishing, and if it is very small, it may lead to gradient exploding. Meanwhile, the network is also sensitive to the changes of $\|\boldsymbol{x}^0\|_2^2$.*

### 3.2.2 PRENORM

**Definition of PreNorm.** PreNorm in Transformer is defined as the normalization applied before the self-attention (or the feed-forward) module. As shown in Figure 2 (b), PreNorm is defined as

$$\boldsymbol{Y} = \text{Self-Attention}(\boldsymbol{X}'), \quad \text{where} \quad \boldsymbol{X}' = [\boldsymbol{x}'_1, \cdots, \boldsymbol{x}'_n], \quad \boldsymbol{x}'_j = \text{PreNorm}(\boldsymbol{x}_j). \quad (4)$$

A single-head self-attention is defined as

$$\boldsymbol{Y} = \boldsymbol{W}_v \boldsymbol{X} \boldsymbol{A},$$

where $\boldsymbol{A} = \text{softmax}(\frac{\boldsymbol{P}}{\sqrt{d_q}}) \in \mathcal{R}^{n \times n}$ is called as the attention matrix, $\boldsymbol{P} = \boldsymbol{X}^\top \boldsymbol{W}_q^\top \boldsymbol{W}_k \boldsymbol{X}$ in which $\frac{\boldsymbol{P}}{\sqrt{d_q}}$ is called as the logit, and $\boldsymbol{X} \in \mathcal{R}^{d \times n}, \boldsymbol{W}_q \in \mathcal{R}^{d_q \times d}, \boldsymbol{W}_k \in \mathcal{R}^{d_q \times d}, \boldsymbol{W}_v \in \mathcal{R}^{d_v \times d}$.

Herein, our goal is to calculate $\frac{\partial \text{vec}(\boldsymbol{Y})}{\partial \text{vec}(\boldsymbol{X})}$. By vectorization of $\boldsymbol{Y} = \boldsymbol{W}_v \boldsymbol{X} \boldsymbol{A}$, we have

$$\partial \text{vec}(\boldsymbol{Y}) = (\boldsymbol{A}^\top \otimes \boldsymbol{W}_v)\partial \text{vec}(\boldsymbol{X}) + (\boldsymbol{I}_n \otimes \boldsymbol{W}_v \boldsymbol{X})\partial \text{vec}(\boldsymbol{A}).$$

Putting together all these terms, we have that

$$\frac{\partial \text{vec}(\boldsymbol{Y})}{\partial \text{vec}(\boldsymbol{X})} = (\boldsymbol{A}^\top \otimes \boldsymbol{W}_v) + (\boldsymbol{I}_n \otimes \boldsymbol{W}_v \boldsymbol{X}) \frac{\boldsymbol{J}}{\sqrt{d_q}} \left( (\boldsymbol{X}^\top \boldsymbol{W}_k^\top \boldsymbol{W}_q \otimes \boldsymbol{I}_n)\boldsymbol{C} + (\boldsymbol{I}_n \otimes \boldsymbol{X}^\top \boldsymbol{W}_q^\top \boldsymbol{W}_k) \right). \quad (5)$$

where $\otimes$ denotes the Kronecker product, $\boldsymbol{C}_{dn}$ is the commutation matrix, and

$$\boldsymbol{J} = \text{blockdiag}\left(\text{diag}(\boldsymbol{A}_{:,1}) - \boldsymbol{A}_{:,1}\boldsymbol{A}_{:,1}^\top, \ldots, \text{diag}(\boldsymbol{A}_{:,n}) - \boldsymbol{A}_{:,n}\boldsymbol{A}_{:,n}^\top\right).$$

The detailed derivation process can also be found in prior work (Qi et al., 2025a). Nevertheless, we note that *rather than analyzing $\boldsymbol{W}_q^\top \boldsymbol{W}_k$ in the self-attention module, here we analyze the influence of $\boldsymbol{X}$*.

According to the Jacobian matrix in Equation 5, we have the following proposition.

**Proposition 2** (PreNorm can stabilize the gradient in self-attention module). *If, for each column $\boldsymbol{x}_j$, we have $\boldsymbol{x}'_j = \alpha_j \boldsymbol{x}_j$ where $\alpha_j \in \mathcal{R}$ is a normalization scalar, according to Equation 4, with the same $\boldsymbol{W}_q, \boldsymbol{W}_k$ and $\boldsymbol{W}_v$, and let $\boldsymbol{Y} = \text{Self-Attention}(\boldsymbol{X})$ and $\boldsymbol{Y}' = \text{Self-Attention}(\boldsymbol{X}')$. Then we have that the Jacobian matrices at $\boldsymbol{X}$ and $\boldsymbol{X}'$ are the same, i.e., $\frac{\partial vec(\boldsymbol{Y})}{\partial vec(\boldsymbol{X})} = \frac{\partial vec(\boldsymbol{Y}')}{\partial vec(\boldsymbol{X}')}$.*

According to Proposition 2, we have the following remark.

**Remark 2.** *PreNorm will guarantee that the norms of the column vectors of $\boldsymbol{X}$ (which are the input to the self-attention layers) are in a relatively stable range. According to Equation 5, we see that if these norms are relatively stable, then the Jacobian matrix will also be stable. Meanwhile, since the gradients with respect to $\boldsymbol{W}_q, \boldsymbol{W}_k$ and $\boldsymbol{W}_v$ are directly involving $\boldsymbol{X}$, a stable $\boldsymbol{X}$ will guarantee that the gradients with respect to $\boldsymbol{W}_q, \boldsymbol{W}_k$ and $\boldsymbol{W}_v$ are relatively stable.*

### 3.2.3 MIDNORM

**Definition of MidNorm.** MidNorm in Transformer is defined as the normalization applied after the self-attention or feed-forward module and meanwhile before the residual shortcut. As shown in Figure 2 (b), MidNorm is defined as

$$\boldsymbol{y} = \text{MidNorm}(\boldsymbol{z}), \quad \text{where} \quad \boldsymbol{z} = \text{FFN}(\boldsymbol{x}; \boldsymbol{W}_1, \boldsymbol{W}_2) = \boldsymbol{W}_2 \text{ReLU}(\boldsymbol{W}_1 \boldsymbol{x}). \quad (6)$$

In self-attention, the matrices $\boldsymbol{W}_v$ and $\boldsymbol{W}_o$ can be viewed as similar function as $\boldsymbol{W}_1$ and $\boldsymbol{W}_2$ in FFN. If we only use a single-head attention, then we have $\boldsymbol{z} = \boldsymbol{W}_o\boldsymbol{W}_v\boldsymbol{x}$.

The Jacobian matrix of an FFN can be computed as: $\boldsymbol{J}_{\boldsymbol{z}}(\boldsymbol{x}) = \frac{\partial \boldsymbol{z}}{\partial \boldsymbol{x}} = \frac{\partial \operatorname{FFN}(\boldsymbol{x};\boldsymbol{W}_1,\boldsymbol{W}_2)}{\partial \boldsymbol{x}} = \boldsymbol{W}_2 \operatorname{diag}\left(\mathbf{1}\left(\boldsymbol{W}_1\boldsymbol{x} > 0\right)\right)\boldsymbol{W}_1$. The Jacobian matrix of an RMSNorm layer is $\frac{\partial \boldsymbol{y}}{\partial \boldsymbol{z}} = \frac{\sqrt{d}}{\|\boldsymbol{z}\|_2} \operatorname{diag}(\boldsymbol{\gamma}) \left(\boldsymbol{I} - \frac{\boldsymbol{z}\boldsymbol{z}^\top}{\|\boldsymbol{z}\|_2^2}\right)$. The Jacobian matrix of an FFN followed by an RMSNorm is:

$$\frac{\partial \boldsymbol{y}}{\partial \boldsymbol{x}} = \frac{\partial \boldsymbol{y}}{\partial \boldsymbol{z}}\frac{\partial \boldsymbol{z}}{\partial \boldsymbol{x}} = \sqrt{d}\operatorname{diag}(\boldsymbol{\gamma})\left(\boldsymbol{I} - \frac{\boldsymbol{z}\boldsymbol{z}^\top}{\|\boldsymbol{z}\|_2^2}\right)\frac{\boldsymbol{W}_2\operatorname{diag}\left(\mathbf{1}\left(\boldsymbol{W}_1\boldsymbol{x} > 0\right)\right)\boldsymbol{W}_1}{\|\boldsymbol{W}_2\operatorname{ReLU}(\boldsymbol{W}_1\boldsymbol{x})\|_2}. \tag{7}$$

**Proposition 3** (Effect of MidNorm). *Let* $\boldsymbol{M} = \frac{\boldsymbol{W}_2\operatorname{diag}\left(\mathbf{1}(\boldsymbol{W}_1\boldsymbol{x}>0)\right)\boldsymbol{W}_1}{\|\boldsymbol{W}_2\operatorname{ReLU}(\boldsymbol{W}_1\boldsymbol{x})\|_2}$*, in a high-dimensional settings when* $\boldsymbol{W}_1$*,* $\boldsymbol{W}_2$ *and* $\boldsymbol{x}$ *are high-dimensional and random, the singular values of* $\boldsymbol{M}$ *will be only related to the shape of* $\boldsymbol{W}_1$ *and* $\boldsymbol{W}_2$*, and will be independent to the magnitude of* $\boldsymbol{W}_1$ *and* $\boldsymbol{W}_2$*.*

According to Proposition 3, we have the following remark.

**Remark 3.** *MidNorm can effectively guarantee that the norms of* $\boldsymbol{W}_1$*,* $\boldsymbol{W}_2$*,* $\boldsymbol{W}_v$*, and* $\boldsymbol{W}_o$ *will not affect the Jacobian matrix as shown in Equation 7. It means that even the magnitudes of these weight matrices are very large, it will not magnify the gradients due to the normalization.*

### 3.2.4 POSTNORM

**Definition of PostNorm.** PostNorm in Transformer is defined as the normalization applied after the residual block. As shown in Figure 2 (d), PostNorm is defined as

$$\boldsymbol{x}^{l+1} = \operatorname{PostNorm}(\boldsymbol{z}^{l+1}), \text{where } \boldsymbol{z}^{l+1} = \boldsymbol{x}^l + f(\boldsymbol{x}^l; \boldsymbol{W}^{l+1}). \tag{8}$$

**Proposition 4** (PostNorm is sensitive to the vector norm of activation). *If* $\boldsymbol{z}^{l+1}$ *in Equation 8 is very large, then it will significantly decrease the gradient.*

*Proof.* From Equation 8, we have $\frac{\partial \boldsymbol{x}^{l+1}}{\partial \boldsymbol{z}^{l+1}} = \frac{\sqrt{d}}{\|\boldsymbol{z}^{l+1}\|_2}\operatorname{diag}(\boldsymbol{\gamma})(\boldsymbol{I} - \frac{\boldsymbol{z}^{l+1}\boldsymbol{z}^{l+1\top}}{\|\boldsymbol{z}^{l+1}\|_2^2})$. If $\|\boldsymbol{z}^{l+1}\|_2$ is very large, according to $\frac{\partial L}{\partial \boldsymbol{z}^{l+1}} = \frac{\partial L}{\partial \boldsymbol{x}^{l+1}}\frac{\partial \boldsymbol{x}^{l+1}}{\partial \boldsymbol{z}^{l+1}}$, we have that the gradient of $\frac{\partial L}{\partial \boldsymbol{z}^{l+1}}$ will be significantly decreased. □

In a classical Transformer (Vaswani et al., 2017), if $f(\boldsymbol{x}; \boldsymbol{W}_1, \boldsymbol{W}_2) = \boldsymbol{W}_2\operatorname{ReLU}(\boldsymbol{W}_1\boldsymbol{x})$, along with the training process, the largest singular values of $\boldsymbol{W}_1$ and $\boldsymbol{W}_2$, *i.e.*, $\sigma_1(\boldsymbol{W}_1)$ and $\sigma_1(\boldsymbol{W}_2)$ will usually become too large (*e.g.*, around 1000). In this way, $\|f(\boldsymbol{x}^l; \boldsymbol{W}^{l+1})\|_2$ will be very large. As a sequence, $\boldsymbol{z}^{l+1}$ in Equation 8 will be very large. Therefore, we have that PostNorm under this circumstance will lead to gradient vanishing.

**Remark 4.** *We need to be very careful when using PostNorm. That is, we must ensure that the norm of the input vector to PostNorm is within a reasonable range; otherwise the network is likely to cause a gradient vanishing when* $\boldsymbol{z}^l$ *being very large or a gradient exploding* $\boldsymbol{z}^l$ *when* $\boldsymbol{z}^l$ *being very small.*

### 3.2.5 QKNORM

**Definition of QKNorm.** QKNorm (Henry et al., 2020) in Transformer is defined as the normalization applied on the queries and keys in the self-attention block. As shown in Figure 2 (e), the self-attention with QKNorm (Dehghani et al., 2023) is defined as

$$\boldsymbol{Y} = \boldsymbol{W}_v\boldsymbol{X}\boldsymbol{A}', \text{where } \boldsymbol{A}' = \operatorname{softmax}(\frac{\boldsymbol{P}'}{\sqrt{d_h}}), \ \boldsymbol{P}' = \boldsymbol{Q}'^\top\boldsymbol{K}', \tag{9}$$

where $\boldsymbol{q}_i'$ and $\boldsymbol{k}_i'$ are the $i$-th column and the $j$-th column in $\boldsymbol{Q}'$ and $\boldsymbol{K}'$, respectively, in which

$$\begin{aligned}
\boldsymbol{q}_i' &= \operatorname{QKNorm}(\boldsymbol{W}_q\boldsymbol{x}_i) = \boldsymbol{\gamma}_q \odot \frac{\sqrt{d_h}\boldsymbol{W}_q\boldsymbol{x}_i}{\|\boldsymbol{W}_q\boldsymbol{x}_i\|_2} = \sqrt{d_h}\operatorname{diag}(\boldsymbol{\gamma}_q)\frac{\boldsymbol{W}_q\boldsymbol{x}_i}{\|\boldsymbol{W}_q\boldsymbol{x}_i\|_2}, \\
\boldsymbol{k}_j' &= \operatorname{QKNorm}(\boldsymbol{W}_k\boldsymbol{x}_j) = \boldsymbol{\gamma}_k \odot \frac{\sqrt{d_h}\boldsymbol{W}_k\boldsymbol{x}_j}{\|\boldsymbol{W}_k\boldsymbol{x}_j\|_2} = \sqrt{d_h}\operatorname{diag}(\boldsymbol{\gamma}_k)\frac{\boldsymbol{W}_k\boldsymbol{x}_j}{\|\boldsymbol{W}_k\boldsymbol{x}_j\|_2},
\end{aligned} \tag{10}$$

where $d_h$ is the head dimension.

To facilitate the derivation, we denote $\boldsymbol{Q} = \boldsymbol{W}_q\boldsymbol{X}$ and $\boldsymbol{K} = \boldsymbol{W}_k\boldsymbol{X}$ as before, and use $\boldsymbol{q}_i$ and $\boldsymbol{k}_j$ to denote the $i$-th column and the $j$-th column in $\boldsymbol{Q}$ and $\boldsymbol{K}$, respectively. Thus, we have that $\boldsymbol{q}'_i = \text{QKNorm}(\boldsymbol{q}_i)$ and $\boldsymbol{k}'_j = \text{QKNorm}(\boldsymbol{k}_j)$. Moreover, we have that $P'_{ij} = \boldsymbol{q}'^\top_i \boldsymbol{k}'_j$, where $P'_{ij}$ is a scalar, and the gradient is computed as follows:

$$
\begin{aligned}
\frac{\partial P'_{ij}}{\partial \boldsymbol{x}} &= \boldsymbol{k}'^\top_j \frac{\partial \boldsymbol{q}'_i}{\partial \boldsymbol{x}} + \boldsymbol{q}'^\top_i \frac{\partial \boldsymbol{k}'_j}{\partial \boldsymbol{x}} \\
&= \sqrt{d_h}\,\text{diag}(\boldsymbol{\gamma}_q)\boldsymbol{k}'^\top_j (\boldsymbol{I} - \frac{\boldsymbol{q}'_i \boldsymbol{q}'_i}{\|\boldsymbol{q}'_i\|^2_2})\frac{\boldsymbol{W}_q}{\|\boldsymbol{W}_q \boldsymbol{x}_i\|_2} + \sqrt{d_h}\,\text{diag}(\boldsymbol{\gamma}_k)\boldsymbol{q}'^\top_i (\boldsymbol{I} - \frac{\boldsymbol{k}'_j \boldsymbol{k}'_j}{\|\boldsymbol{k}'_j\|^2_2})\frac{\boldsymbol{W}_k}{\|\boldsymbol{W}_k \boldsymbol{x}_i\|_2}.
\end{aligned}
\tag{11}
$$

**Proposition 5** (Effect of QKNorm). *In a high-dimensional settings, i.e., when all $\boldsymbol{W}_q$, $\boldsymbol{W}_k$ and $\boldsymbol{x}$ are high-dimensional and random, in Equation 11, the gradient term of $\frac{\partial P'_{ij}}{\partial \boldsymbol{x}}$ is independent of the magnitudes $\boldsymbol{W}_q$ and $\boldsymbol{W}_k$.*

According to the Proposition, we have the following remark.

**Remark 5.** *QKNorm can mitigate the joint effect of $\boldsymbol{W}_q^\top \boldsymbol{W}_k$ on the gradient of the self-attention layer. The fast increase of the singular values of $\boldsymbol{W}_q^\top \boldsymbol{W}_k$ has been revealed to be a root reason leading to model crash. Our analysis shows that QKNorm can effectively mitigate the reason to cause model crash brought by $\boldsymbol{W}_q^\top \boldsymbol{W}_k$.*

Though QKNorm can mitigate the problem brought by $\boldsymbol{W}_q^\top \boldsymbol{W}_k$, it cannot fully replace the role of PreNorm, because PreNorm can jointly deal with the problem of $\boldsymbol{W}_q$, $\boldsymbol{W}_k$ and $\boldsymbol{W}_v$, and the gradient of $\boldsymbol{W}_v$ is also affected by the value of $\boldsymbol{X}$.

## 3.3 DNT: A Transformer that can relieve the issue of heavy-tail gradients

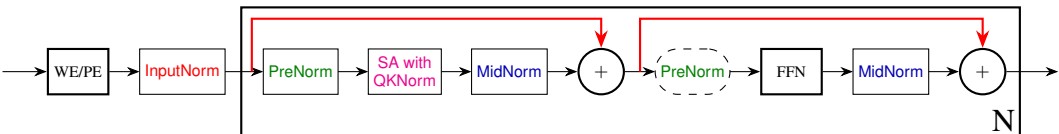

FIGURE 3: DNT architecture. The second PreNorm marked with dashed and rounded corners is optional. By default, we do not use the second PreNorm.

Having analyzed the effects of different normalizations, we use four types of normalizations, including InputNorm, PreNorm, MidNorm and QKNorm, except for PostNorm. *The reason why we do not use PostNorm is that it may cause some training instability issue.* For clarity, we illustrate the architecture of our DNT in Figure 3.

Our DNT model commences with Word Embeddings (WE) or Patch Encodings (PE). Then, the initial representations undergo an InputNorm processing and the normalized embeddings are used for the subsequent operations. The core transformer module consists of $N$ blocks. In each block, a PreNorm is applied at first, followed by a self-attention which is augmented with query-key normalization (*i.e.*, QKNorm). Then, a MidNorm is applied to process the attention outputs before integrating with the residual connection. In the second sub-block, before the feed-forward network (FFN), the PreNorm is optional; and after FFN, a MidNorm is applied before integrating with the residual connection. This entire structure is replicated $N$ times to form the whole architecture.

We visualize the effects of each normalization in our DNT in Figure 4. According to the analysis mentioned above, we summarize the advantages of our DNT as following: a) the magnitude of $\boldsymbol{x}^0$ will significantly affect the gradient of each layer in the Transformer, but we introduce InputNorm to resolve the influence of $\boldsymbol{x}^0$; b) PreNorm can constrain the norm of each column in activations $\boldsymbol{X}$ in each timestep, and thus amend the Jacobian matrix of self-attention to not be significantly affected by the magnitude of $\boldsymbol{X}$; c) MidNorm will amend the Jacobian matrix of each sub-block (*i.e.*, the sub-block with self-attention and the sub-block with FFN) in our DNT to not be affected by

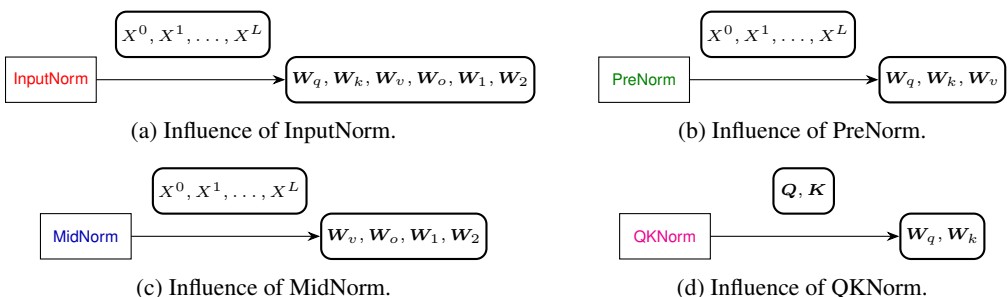

(a) Influence of InputNorm.

(b) Influence of PreNorm.

(c) Influence of MidNorm.

(d) Influence of QKNorm.

FIGURE 4: Influence of different normalizations. For instance, InputNorm stabilizes $\boldsymbol{W}_q, \boldsymbol{W}_k, \boldsymbol{W}_v, \boldsymbol{W}_o, \boldsymbol{W}_1, \boldsymbol{W}_2$ by constraining $X^0, X^1, \ldots, X^L$.

the magnitude of $\boldsymbol{W}_1, \boldsymbol{W}_2, \boldsymbol{W}_v$ and $\boldsymbol{W}_o$; d) QKNorm can relieve or even remove the influence of the magnitude of $\boldsymbol{W}_q$ and $\boldsymbol{W}_k$ on the Jacobian matrix of self-attention, and thus reduce the risk of problems, such as rank collapse (Noci et al., 2022), entropy collapse (Zhai et al., 2023), or spectral energy concentration (Qi et al., 2025a) caused by $\boldsymbol{W}_q^\top \boldsymbol{W}_k$.

In DNT, we use four different types of normalizations. We observe that nGPT (Loshchilov et al., 2025) also uses some of the normalizations mentioned above. Here, we would like to emphasize the differences between DNT and nGPT that: a) DNT provides theoretical justifications for each normalization in different position; b) DNT uses InputNorm rather than PostNorm, whereas nGPT use many PostNorms but not InputNorm; c) nGPT normalizes the activations or the weights into spheres, whereas DNT only normalizes the activations but does not requires activations on spheres.

We term our model as Deeply Normalized Transformer (DNT for short), because it is designed by properly adding or positioning normalization operators in the conventional Transformer. For vision problem, we term it as V-DNT, and for language problem, we term it as L-DNT. The key difference between V-DNT and L-DNT is that V-DNT uses patch embedding, but L-DNT uses word embedding and mask for attention computation.

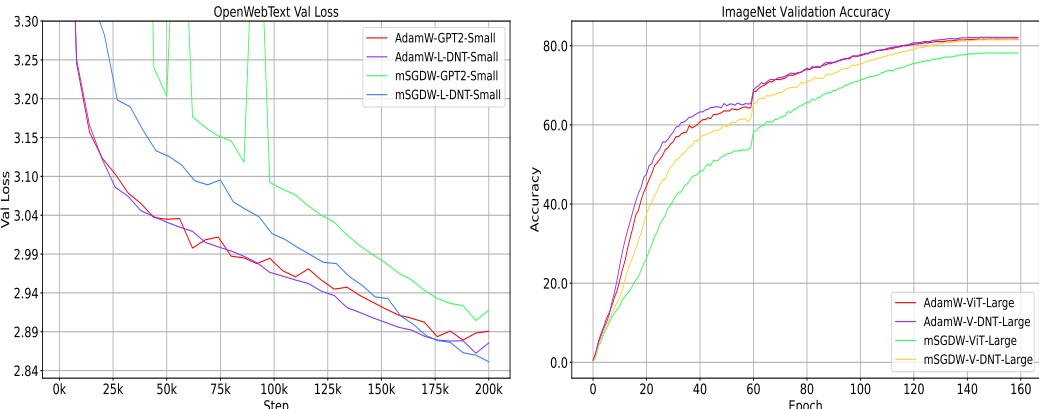

FIGURE 5: Validation loss (Left) on OpenWebText and recognition accuracy (Right) on ImageNet. We compare L-DNT-Small (124M) to GPT2-Small (124M), and V-DNT-Large (307M) to ViT-Large (307M). By effectively relieving the heavy-tail gradient issue, our DNT network trained with naive mSGDW can achieve competitive performance to AdamW (Val loss *2.849 vs. 2.863* on OpenWebText, Acc *81. 5% vs. 82. 1%* on ImageNet). However, in classical Transformer with PreNorm, the performance of mSGDW under-performs AdamW significantly (Val loss *2.906 vs 2.867* on OpenWebText, Acc *78.2% vs 81.7%* on ImageNet). See Appendix C for the training parameters.

## 4 EXPERIMENTS

We conduct experiments with two popular Transformer architectures: Vision Transformer (ViT) and Generative Pretrained Transformer (GPT). Our implementation leverages established repositories:

timm (Wightman, 2019) for ViT and nanoGPT (Karpathy, 2022) for GPT models. For experiments with ViT, we utilized two model scales: ViT-Large (307M parameters) and ViT-Huge (632M), following the configurations described in (Dosovitskiy et al., 2020). The data augmentation strategy aligns with (Xie et al., 2024) to ensure fair comparison with previously reported results. For experiments with GPT, we employed the nanoGPT implementation focusing on GPT2-Small (124M) and GPT2-Large (774M) variants due to computational constraints. The results of our baselines align with previous work, including Sophia (Liu et al., 2023b) on OpenWebText and MAE (He et al., 2022) on ImageNet. Training was conducted using PyTorch (Paszke et al., 2019) with bfloat16 precision on A800 GPUs, employing a cosine learning rate schedule.

## 4.1 VISUALIZATION OF GRADIENTS OF DNT AND TRANSFORMER WITH PRENORM

To visually compare the standard Transformer with our DNT, we visualize the gradients with respect to different weights, including $W_q, W_k, W_v, W_o, W_1$ and $W_2$. We chose the early checkpoints of the model training for visualization, but we found that the same phenomenon also occurs in the middle and later stages of the model training. The visualization results are shown in Figure 1. We can see that our DNT network can well relieve the issue of heavy tail distribution of the gradients. For instance, in the Transformer, the amplitude of the entries in the gradients almost spreads in the range $[0, 10^{-4}]$; whereas the amplitude of the entries in the gradients in our DNT concentrates around $[0, 10^{-5}]$.

## 4.2 MSGDW ACHIEVES PERFORMANCE ON PAR WITH ADAMW.

We also give a quantitative comparison of the standard Transformer and our DNT both trained with Adam and mSGD on OpenWebText and ImageNet in Table 1. We can see that training our DNT via mSGDW achieves a similar result to that is trained with AdamW. We can also see that using mSGDW to train our DNT model greatly outperforms the performance of using mSGDW to train the standard Transformer. In Figure 5, we display the validation loss on OpenWebText and the training accuracy on ImageNet along with the training process. Note that we did not tune the learning rate too much. We just followed the learning rate settings in the previous works Karpathy (2022); Liu et al. (2023b). We believe tuning learning rate will bring in some differences. But overall, DNT network can enable mSGDW compete with AdamW.

TABLE 1: Quantitative comparison of standard ViT/GPT2 and V-DNT/L-DNT trained with AdamW and mSGDW on OpenWebText and ImageNet. Results on ImageNet are based on 150 epochs, and results on OpenWebText are based on 200K steps.

| Optimizer | Types of Model | ImageNet (Acc. ↑) | | OpenWebText (Val Loss. ↓) | | |
|---|---|---|---|---|---|---|
| | | 307M | 632M | 124M | 774M | 1436M |
| AdamW | ViT/GPT2 | 81.7 | 80.8 | 2.867 | 2.492 | 2.435 |
| AdamW | V-DNT/L-DNT | **82.1** | **81.9** | **2.863** | **2.481** | **2.396** |
| mSGDW | ViT/GPT2 | 78.2 | 73.5 | 2.906 | 2.544 | 2.472 |
| mSGDW | V-DNT/L-DNT | **81.5** | **81.2** | **2.849** | **2.503** | **2.408** |

## 4.3 HOW MUCH MEMORY MSGDW SAVES RATHER THAN ADAMW?

We compare the memory usage by mSGDW and AdamW. The results are shown in Table 2. Theoretically, we can calculate that the memory taken by AdamW (only the optimizer part) is 11.5GB, and the memory costed by mSGDW (only the optimizer part) is 5.7GB. In the experiment, we obtained DNT+AdamW (model plus optimizer) costs 67GB, and DNT+mSGDW (model plus optimizer) costs 61GB. Using mSGDW instead of AdamW on 1.4B model can save around 6GB memory.

TABLE 2: Comparision of GPU memory used by mSGDW and AdamW trained on 1.4B DNT model. DNT+AdamW means the network usage and the optimizer usage of GPU memory. † denotes the theoretically calculated values, and ‡ denotes the observed values in practice.

|  | AdamW | mSGDW | DNT+AdamW | DNT+mSGDW |
| --- | --- | --- | --- | --- |
| Memory | $11.5^\dagger$ GB | $5.7^\dagger$ GB | $\approx 67^\ddagger$ GB | $\approx 61^\ddagger$ GB |

## 4.4 ABLATION STUDY

**Comparison of different normalization methods.** We conduct ablation study of five different normalization methods. Figure 7 in the Appendix D illustrates these five different network settings. Let us brief introduce these five settings below: 1) Setting 1: Standard transformer with PreNorm, for which we abbreviate it as S1; 2) Setting 2: S1 + QKNorm; 3) Setting 3: S2 + InputNorm; 4) Setting 4: 2PreNorms + MidNorm + QKNorm + InputNorm; 5) Setting 5: only 1 PreNorm before self-attention + MidNorm + QKNorm + InputNorm. We use momentum mSGDW for all the training in this subsection. All models are trained with the same hyper-parameters. The results are shown in Figure 6. We have the following observations.

- On OpenWebText, the original PreNorm setting (S1) shows the worst performance. The performance of S2 is similar to that of S1. The setting of S3 with input obtained better performance. Finally, S4 and S5 obtained the best performance. Meanwhile, the performance of S4 and S5 is similar.
- On ImageNet, the original PreNorm setting (S1) is significantly worse than the other four settings. S3 achieves the best performance, and S4 and S5 also obtain excellent performance.

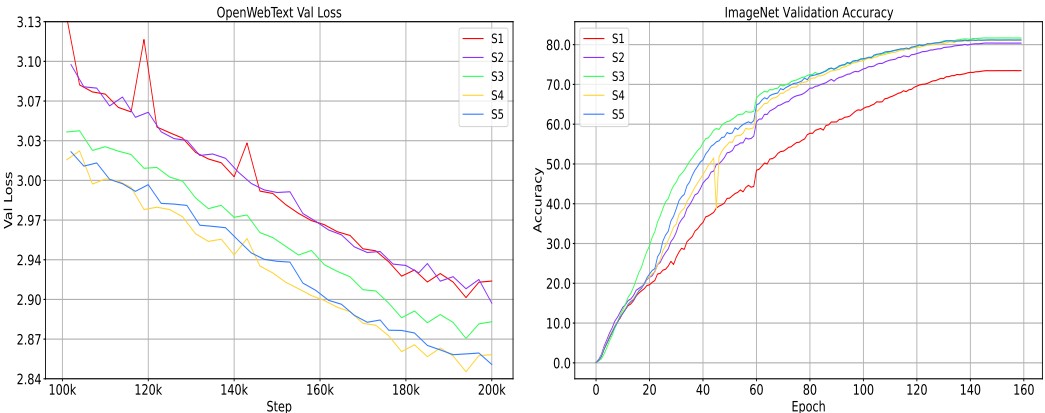

FIGURE 6: Ablation study of different settings using mSGD optimizer on ImageNet and OpenWeb-Text. Left side shows accuracy curve of Huge vision model (632M) on ImageNet. Right side shows the validation loss of language model (124M) on OpenWebText.

## 5 CONCLUSION

We introduced a novel architecture, named Deeply Normalized Transformer (DNT), which enables efficient training with the vanilla momentum SGDW (mSGDW), achieving performance on par with AdamW-optimized Transformers. Unlike traditional approaches that rely on sophisticated optimizers to address the challenges of heavy-tailed gradient distributions, our DNT properly integrated normalization techniques into the architecture of Transformer to effectively regulate the Jacobian matrices of each block, the contributions of the weights, activations, and their interactions, and thus make the gradient distribution concentrated. Our findings demonstrated that a properly designed architecture can make a simple optimizers like mSGDW just as effective as sophisticated ones. This opened new opportunities for creating more efficient, scalable, and accessible Transformer models.

ACKNOWLEDGMENTS

Zhouchen Lin is supported by the National Natural Science Foundation of China (NSFC) under Grant No. 62276004. Chun-Guang Li is supported by NSFC under Grant No. 62576048.

## ETHICS STATEMENT

This work presents a novel deeply normalized Transformer architecture. It does not involve human subjects and poses no potential risks. The study is free from conflicts of interest, sponsorship issues, or concerns related to discrimination, bias, or fairness. All data used adhere to legal and ethical standards, and privacy and security considerations have been addressed. Our work fully adheres to research integrity principles, and no ethical concerns have arisen during the course of this study.

## REPRODUCIBILITY STATEMENT

To facilitate reproducibility, we provide comprehensive experimental details in the Appendices, including theoretical proofs, experimental settings, and configurations. Our implementation builds on nanoGPT and timm. The ImageNet and OpenWebText datasets are publicly available.

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

## A    LLM USAGE

During the preparation of this work, the authors used ChatGPT for language editing and to assist in the creation of TikZ diagrams. The models were firstly prompted with draft text or rough sketches to improve clarity and fluency of language and to generate code snippets for figures. Then, we carefully reviewed and modified all generated content. The core ideas, research, analysis, and conclusions remain entirely the work of the authors, and the LLMs were not involved in any intellectual contribution.

## B    PROOF OF PROPOSITION 2, 3 AND 5

Proposition 1 and Proposition 4 are very easy to prove, we have given a brief proof in the main body. Therefore, in the appendix, we only provide the proof of Proposition 2, 3 and 5.

### B.1    PROOF OF PROPOSITION 2 ON PRENORM

*Proof.* If for each column $\boldsymbol{x}_j$, we have $\boldsymbol{x}'_j = \alpha_j \boldsymbol{x}_j$, with the same $\boldsymbol{W}_q, \boldsymbol{W}_k$ and $\boldsymbol{W}_v$, we have that $\boldsymbol{Y} = \boldsymbol{Y}'$ given $\boldsymbol{Y} = \text{Self-Attention}(\boldsymbol{X})$ and $\boldsymbol{Y}' = \text{Self-Attention}(\boldsymbol{X}')$ because we will obtain the same input to the self-attention after the PreNorm.

That is, after the PreNorm, we have that $\text{PreNorm}(\boldsymbol{X}) = \text{PreNorm}(\boldsymbol{X}')$, according to Equation 5, thun we have that: $\frac{\partial \text{vec}(\boldsymbol{Y})}{\partial \text{vec}(\boldsymbol{X})} = \frac{\partial \text{vec}(\boldsymbol{Y}')}{\partial \text{vec}(\boldsymbol{X}')}$.

□

A single-head self-attention can be defined as

$$\boldsymbol{Y} = \boldsymbol{W}_v \boldsymbol{X} \boldsymbol{A},$$

where $\boldsymbol{A} = \text{softmax}(\frac{\boldsymbol{P}}{\sqrt{d_q}})$ and $\boldsymbol{P} = \boldsymbol{X}^\top \boldsymbol{W}_q^\top \boldsymbol{W}_k \boldsymbol{X}$, in which $\boldsymbol{A}$ is called as the attention matrix and $\frac{\boldsymbol{P}}{\sqrt{d_q}}$ is called as the logit, and $\boldsymbol{A} \in \mathcal{R}^{n \times n}, \boldsymbol{X} \in \mathcal{R}^{d \times n}, \boldsymbol{W}_v \in \mathcal{R}^{d_v \times d}$. Here, our goal is to calculate $\frac{\partial \text{vec}(\boldsymbol{Y})}{\partial \text{vec}(\boldsymbol{X})}$. By vectorization of $\boldsymbol{Y} = \boldsymbol{W}_v \boldsymbol{X} \boldsymbol{A}$, we have

$$\partial \text{vec}(\boldsymbol{Y}) = (\boldsymbol{A}^\top \otimes \boldsymbol{W}_v)\partial \text{vec}(\boldsymbol{X}) + (\boldsymbol{I}_n \otimes \boldsymbol{W}_v \boldsymbol{X})\partial \text{vec}(\boldsymbol{A}).$$

Bringing in all the terms, we get the following formula

$$\frac{\partial \text{vec}(\boldsymbol{Y})}{\partial \text{vec}(\boldsymbol{X})} = (\boldsymbol{A}^\top \otimes \boldsymbol{W}_v) + (\boldsymbol{I}_n \otimes \boldsymbol{W}_v \boldsymbol{X})\frac{\boldsymbol{J}}{\sqrt{d_q}}\left( (\boldsymbol{X}^\top \boldsymbol{W}_k^\top \boldsymbol{W}_q \otimes \boldsymbol{I}_n)\boldsymbol{C}_{dn} + (\boldsymbol{I}_n \otimes \boldsymbol{X}^\top \boldsymbol{W}_q^\top \boldsymbol{W}_k) \right),$$

where $\boldsymbol{C}_{dn}$ is the commutation matrix  and

$$\boldsymbol{J} = \text{blockdiag}(\text{diag}(\boldsymbol{A}_{:,1}) - \boldsymbol{A}_{:,1}\boldsymbol{A}_{:,1}^\top, \ldots, \text{diag}(\boldsymbol{A}_{:,n}) - \boldsymbol{A}_{:,n}\boldsymbol{A}_{:,n}^\top).$$

Note that $\boldsymbol{J}$ is a function of $\boldsymbol{A}$, and $\boldsymbol{A}$ is a function of $\boldsymbol{X}$ associated with softmax function. Obviously, $\frac{\partial \text{vec}(\boldsymbol{Y})}{\partial \text{vec}(\boldsymbol{X})}$ is a high-order function of $\boldsymbol{X}$ where the softmax makes the analysis more complicated.

Here, we drop the softmax operation and give an analysis of the Jacobian matrix of the linear attention module, *i.e.*, $\boldsymbol{A} = \frac{\boldsymbol{P}}{\sqrt{d_q}}$ and $\boldsymbol{P} = \boldsymbol{X}^\top \boldsymbol{W}_q^\top \boldsymbol{W}_k \boldsymbol{X}$. For the linear attention, the Jacobian matrix is:

$$\frac{\partial \text{vec}(\boldsymbol{Y})}{\partial \text{vec}(\boldsymbol{X})} = (\boldsymbol{A}^\top \otimes \boldsymbol{W}_v) + \frac{(\boldsymbol{I}_n \otimes \boldsymbol{W}_v \boldsymbol{X})}{\sqrt{d_q}}\left( (\boldsymbol{X}^\top \boldsymbol{W}_k^\top \boldsymbol{W}_q \otimes \boldsymbol{I}_n)\boldsymbol{C} + (\boldsymbol{I}_n \otimes \boldsymbol{X}^\top \boldsymbol{W}_q^\top \boldsymbol{W}_k) \right).$$

(12)

Obviously, if the norm of each feature vector for each token is large, the magnitude of each element in $\frac{\partial \text{vec}(\boldsymbol{Y})}{\partial \text{vec}(\boldsymbol{X})}$ will have large probability to be large, and the singular value of $\frac{\partial \text{vec}(\boldsymbol{Y})}{\partial \text{vec}(\boldsymbol{X})}$ may be magnified second-orderly by the norm of each column in $\boldsymbol{X}$.

---

https://en.wikipedia.org/wiki/Commutation_matrix

Furthermore, we would like to conduct a deeper analysis of the gradient of the loss with respect to the weights. In the backpropagation, since that we have obtained $\frac{\partial \mathcal{L}}{\partial \text{vec}(\boldsymbol{Y})}$, we would like to further analyze $\frac{\partial \mathcal{L}}{\partial \text{vec}(\boldsymbol{W}_q)}$, $\frac{\partial \mathcal{L}}{\partial \text{vec}(\boldsymbol{W}_k)}$, and $\frac{\partial \mathcal{L}}{\partial \text{vec}(\boldsymbol{W}_v)}$.

For the weight matrix $\boldsymbol{W}_q$, we have

$$
\begin{aligned}
\frac{\partial \mathcal{L}}{\partial \text{vec}(\boldsymbol{W}_q)} &= \frac{\partial \mathcal{L}}{\partial \text{vec}(\boldsymbol{Y})} \frac{\partial \text{vec}(\boldsymbol{Y})}{\partial \text{vec}(\boldsymbol{A})} \frac{\partial \text{vec}(\boldsymbol{A})}{\partial \text{vec}(\boldsymbol{P})} \frac{\partial \text{vec}(\boldsymbol{P})}{\partial \text{vec}(\boldsymbol{W}_q)}, \\
&= \frac{\partial \mathcal{L}}{\partial \text{vec}(\boldsymbol{Y})} \left(\boldsymbol{I}_n \otimes \boldsymbol{W}_v \boldsymbol{X}\right) \frac{\boldsymbol{J}}{\sqrt{d_q}} \left((\boldsymbol{W}_k \boldsymbol{X})^\top \otimes \boldsymbol{X}^\top\right) \boldsymbol{C}.
\end{aligned}
\tag{13}
$$

For the weight matrix $\boldsymbol{W}_k$, we have

$$
\begin{aligned}
\frac{\partial \mathcal{L}}{\partial \text{vec}(\boldsymbol{W}_k)} &= \frac{\partial \mathcal{L}}{\partial \text{vec}(\boldsymbol{Y})} \frac{\partial \text{vec}(\boldsymbol{Y})}{\partial \text{vec}(\boldsymbol{A})} \frac{\partial \text{vec}(\boldsymbol{A})}{\partial \text{vec}(\boldsymbol{P})} \frac{\partial \text{vec}(\boldsymbol{P})}{\partial \text{vec}(\boldsymbol{W}_k)}, \\
&= \frac{\partial \mathcal{L}}{\partial \text{vec}(\boldsymbol{Y})} \left(\boldsymbol{I}_n \otimes \boldsymbol{W}_v \boldsymbol{X}\right) \frac{\boldsymbol{J}}{\sqrt{d_q}} \left(\boldsymbol{X}^\top \otimes (\boldsymbol{W}_q \boldsymbol{X})^\top\right).
\end{aligned}
\tag{14}
$$

For the weight matrix $\boldsymbol{W}_v$, we know that $\text{vec}(\boldsymbol{Y}) = \left((\boldsymbol{X}\boldsymbol{A})^\top \otimes \boldsymbol{I}\right) \text{vec}(\boldsymbol{W}_v)$, thus we have

$$
\begin{aligned}
\frac{\partial \mathcal{L}}{\partial \text{vec}(\boldsymbol{W}_v)} &= \frac{\partial \mathcal{L}}{\partial \text{vec}(\boldsymbol{Y})} \frac{\partial \text{vec}(\boldsymbol{Y})}{\partial \text{vec}(\boldsymbol{W}_v)}, \\
&= \frac{\partial \mathcal{L}}{\partial \text{vec}(\boldsymbol{Y})} \left((\boldsymbol{A}^\top \boldsymbol{X}^\top) \otimes \boldsymbol{I}\right).
\end{aligned}
\tag{15}
$$

We can see that in Equations 13-15, the gradients of loss with respect to $\boldsymbol{W}_q, \boldsymbol{W}_k$ and $\boldsymbol{W}_v$ are all related to $\boldsymbol{X}$. After PreNorm, $\boldsymbol{X}$ is in a relatively stable range, thus the gradients of $\boldsymbol{W}_q, \boldsymbol{W}_k$ and $\boldsymbol{W}_v$ are all in a stable range.

In conclusion, the normalization of $\boldsymbol{X}$ can help stablize the gradient of the loss function with respect to $\boldsymbol{W}_q, \boldsymbol{W}_k$ and $\boldsymbol{W}_v$, and meanwhile help make $\frac{\partial \text{vec}(\boldsymbol{Y})}{\partial \text{vec}(\boldsymbol{X})}$ more stable.

## B.2 PROOF OF PROPOSITION 3 ON MIDNORM

*Proof.* Starting with the definition $\boldsymbol{W}_1 = \frac{\boldsymbol{W}}{\|\boldsymbol{y}\|_2}$ where $\boldsymbol{y} = \boldsymbol{W}\boldsymbol{x}$ and $\boldsymbol{W} \in \mathcal{R}^{m \times n}$, let's derive the relationship between singular values:

For a random matrix $\boldsymbol{W}$ with i.i.d. entries (*i.e.*, zero mean, variance $\sigma_W^2$) and a random vector $\boldsymbol{x}$ with i.i.d. entries (*i.e.*, zero mean, variance $\sigma_x^2$):

$$
\mathbb{E}[\|\boldsymbol{y}\|_2^2] = \mathbb{E}[\|\boldsymbol{W}\boldsymbol{x}\|_2^2] = \mathbb{E}[\boldsymbol{x}^T \boldsymbol{W}^T \boldsymbol{W} \boldsymbol{x}].
$$

Using the trace property:

$$
\mathbb{E}[\boldsymbol{x}^T \boldsymbol{W}^T \boldsymbol{W} \boldsymbol{x}] = \mathbb{E}[\text{tr}(\boldsymbol{x}^T \boldsymbol{W}^T \boldsymbol{W} \boldsymbol{x})] = \mathbb{E}[\text{tr}(\boldsymbol{W} \boldsymbol{x} \boldsymbol{x}^T \boldsymbol{W}^T)].
$$

With $\boldsymbol{x}$ and $\boldsymbol{W}$ independent, and $\mathbb{E}[\boldsymbol{x}\boldsymbol{x}^T] = \sigma_x^2 \boldsymbol{I}_n$:

$$
\mathbb{E}[\text{tr}(\boldsymbol{W} \boldsymbol{x} \boldsymbol{x}^T \boldsymbol{W}^T)] = \sigma_x^2 \boldsymbol{I}_n \mathbb{E}[\text{tr}(\boldsymbol{W}^T \boldsymbol{W})] = \sigma_x^2 \mathbb{E}[\text{tr}(\boldsymbol{W} \boldsymbol{W}^T)]
$$

For $\boldsymbol{W}$ with i.i.d. entries, $\mathbb{E}[\text{tr}(\boldsymbol{W}\boldsymbol{W}^T)] = \mathbb{E}[\|\boldsymbol{W}\|_F^2] = m \cdot n \cdot \sigma_W^2$.

Therefore, we have:

$$
\mathbb{E}[\|\boldsymbol{y}\|_2^2] = \sigma_x^2 \cdot m \cdot n \cdot \sigma_W^2 = m \cdot n \cdot \sigma_W^2 \cdot \sigma_x^2.
$$

According to the measure concentration property, we have that $\|\boldsymbol{y}\|_2^2$ concentrates around its expectation with high probability:

$$\|\boldsymbol{y}\|_2^2 \approx \mathbb{E}[\|\boldsymbol{y}\|_2^2] = m \cdot n \cdot \sigma_W^2 \cdot \sigma_x^2.$$

By taking the square root, we have:

$$\|\boldsymbol{y}\|_2 \approx \sqrt{m \cdot n} \cdot \sigma_W \cdot \sigma_x \text{ with high probability.}$$

Considering the SVD of $\boldsymbol{W}$, say $\boldsymbol{W} = \boldsymbol{U}\boldsymbol{\Sigma}\boldsymbol{V}^T$, where $\boldsymbol{\Sigma}$ contains singular values $\sigma_i(\boldsymbol{W})$. We have that the singular values of $\boldsymbol{W}_1$ are:

$$\sigma_i(\boldsymbol{W}_1) = \sigma_i\left(\frac{\boldsymbol{W}}{\|\boldsymbol{y}\|_2}\right) = \frac{\sigma_i(\boldsymbol{W})}{\|\boldsymbol{y}\|_2}.$$

Substituting it into our concentration result, we have that:

$$\sigma_i(\boldsymbol{W}_1) \approx \frac{\sigma_i(\boldsymbol{W})}{\sqrt{m \cdot n} \cdot \sigma_W \cdot \sigma_x}.$$

For large random matrices with i.i.d. entries, the random matrix theory (Horn & Johnson, 2012; Tao, 2012) tells us that the largest singular value follows:

$$\sigma_1(\boldsymbol{W}) \approx (\sqrt{m} + \sqrt{n})\sigma_W.$$

Substituting this into our result above, we have:

$$\sigma_1(\boldsymbol{W}_1) \approx \frac{(\sqrt{m} + \sqrt{n})\sigma_W}{\sqrt{m \cdot n} \cdot \sigma_W \cdot \sigma_x} = \frac{\sqrt{m} + \sqrt{n}}{\sqrt{m \cdot n} \cdot \sigma_x}. \tag{16}$$

This derivation result in Equation 16 shows that in high dimension, the largest singular value of $\boldsymbol{W}_1$ becomes essentially deterministic, depending only on the dimensions of $\boldsymbol{W}$ and the statistical property $\sigma_x$ (which is the standard variance of each entry in $\boldsymbol{x}$) of the random vector $\boldsymbol{x}$. If $m = n$, then we have that $\sigma_1(\boldsymbol{W}_1) \approx \frac{2}{\sqrt{m} \cdot \sigma_x}$. □

### B.3 Proof of Proposition 5 on QKNorm

*Proof.* Self-attention with QKNorm (Dehghani et al., 2023) is defined as:

$$\boldsymbol{Y} = \boldsymbol{W}_v \boldsymbol{X} \boldsymbol{A},$$

where $\boldsymbol{A}' = \text{softmax}(\frac{\boldsymbol{P}'}{\sqrt{d_h}})$, $\boldsymbol{P}' = \boldsymbol{Q}'^\top \boldsymbol{K}'$, and $\boldsymbol{q}_i'$ and $\boldsymbol{k}_i'$ are the $i$-th column and the $j$-th column in $\boldsymbol{Q}'$ and $\boldsymbol{K}'$ respectively, and we define

$$\boldsymbol{q}_i' = \text{RMSN}(\boldsymbol{W}_q \boldsymbol{x}_i) = \boldsymbol{\gamma}_q \odot \frac{\sqrt{d_h}\boldsymbol{W}_q \boldsymbol{x}_i}{\|\boldsymbol{W}_q \boldsymbol{x}_i\|_2} = \sqrt{d_h}\,\text{diag}(\boldsymbol{\gamma}_q)\frac{\boldsymbol{W}_q \boldsymbol{x}_i}{\|\boldsymbol{W}_q \boldsymbol{x}_i\|_2},$$

$$\boldsymbol{k}_j' = \text{RMSN}(\boldsymbol{W}_k \boldsymbol{x}_j) = \boldsymbol{\gamma}_k \odot \frac{\sqrt{d_h}\boldsymbol{W}_k \boldsymbol{x}_j}{\|\boldsymbol{W}_k \boldsymbol{x}_j\|_2} = \sqrt{d_h}\,\text{diag}(\boldsymbol{\gamma}_k)\frac{\boldsymbol{W}_k \boldsymbol{x}_j}{\|\boldsymbol{W}_k \boldsymbol{x}_j\|_2}.$$

To facilitate our derivation, we will use $\boldsymbol{Q} = \boldsymbol{W}_q \boldsymbol{X}$ and $\boldsymbol{K} = \boldsymbol{W}_k \boldsymbol{X}$ as before, we use $\boldsymbol{q}_i$ and $\boldsymbol{k}_j$ to denote the $i$-th column and the $j$-th column in $\boldsymbol{Q}$ and $\boldsymbol{K}$, respectively. Thus, we denote $\boldsymbol{q}_i' = \text{RMSN}(\boldsymbol{q}_i)$ and $\boldsymbol{k}_j' = \text{RMSN}(\boldsymbol{k}_j)$.

Therefore, according to the product rule and chain rule, we can denote the Jacobian matrix of $\boldsymbol{Y}$ with respect to $\boldsymbol{X}$ as follows:

$$
\begin{aligned}
\frac{\partial \text{vec}(\boldsymbol{Y})}{\partial \text{vec}(\boldsymbol{X})} &= (\boldsymbol{A'}^{\top} \otimes \boldsymbol{W}_v) + (\boldsymbol{I}_n \otimes \boldsymbol{W}_v \boldsymbol{X}) \frac{\partial \text{vec}(\boldsymbol{A'})}{\partial \text{vec}(\boldsymbol{X})} \\
&= (\boldsymbol{A'}^{\top} \otimes \boldsymbol{W}_v) + (\boldsymbol{I}_n \otimes \boldsymbol{W}_v \boldsymbol{X}) \frac{\partial \text{vec}(\boldsymbol{A'})}{\partial \text{vec}(\boldsymbol{P'})} \frac{\partial \text{vec}(\boldsymbol{P'})}{\partial \text{vec}(\boldsymbol{X})} \\
&= (\boldsymbol{A'}^{\top} \otimes \boldsymbol{W}_v) + (\boldsymbol{I}_n \otimes \boldsymbol{W}_v \boldsymbol{X}) \frac{\partial \text{vec}(\boldsymbol{A'})}{\partial \text{vec}(\boldsymbol{P'})} \left( \frac{\partial \text{vec}(\boldsymbol{P'})}{\partial \text{vec}(\boldsymbol{Q'})} \frac{\partial \text{vec}(\boldsymbol{Q'})}{\partial \text{vec}(\boldsymbol{X'})} + \frac{\partial \text{vec}(\boldsymbol{P'})}{\partial \text{vec}(\boldsymbol{K'})} \frac{\partial \text{vec}(\boldsymbol{K'})}{\partial \text{vec}(\boldsymbol{X'})} \right) \\
&= (\boldsymbol{A'}^{\top} \otimes \boldsymbol{W}_v) + (\boldsymbol{I}_n \otimes \boldsymbol{W}_v \boldsymbol{X}) \frac{\partial \text{vec}(\boldsymbol{A'})}{\partial \text{vec}(\boldsymbol{P'})} \left( \frac{\partial \text{vec}(\boldsymbol{P'})}{\partial \text{vec}(\boldsymbol{Q'})} \frac{\partial \text{vec}(\boldsymbol{Q'})}{\partial \text{vec}(\boldsymbol{Q})} \frac{\partial \text{vec}(\boldsymbol{Q})}{\partial \text{vec}(\boldsymbol{X})} + \frac{\partial \text{vec}(\boldsymbol{P'})}{\partial \text{vec}(\boldsymbol{K'})} \frac{\partial \text{vec}(\boldsymbol{K'})}{\partial \text{vec}(\boldsymbol{K})} \frac{\partial \text{vec}(\boldsymbol{K})}{\partial \text{vec}(\boldsymbol{X})} \right)
\end{aligned}
$$

To derive out $\frac{\partial \text{vec}(\boldsymbol{Y})}{\partial \text{vec}(\boldsymbol{X})}$, we need to derive out each term in the above equation.

Since we know $\boldsymbol{P'} = \boldsymbol{Q'}^{\top} \boldsymbol{K'}$, then we have,

$$
\frac{\partial \text{vec}(\boldsymbol{P'})}{\partial \text{vec}(\boldsymbol{K'})} = \boldsymbol{I} \otimes \boldsymbol{Q'}^{\top}.
$$

Similarly, we have

$$
\frac{\partial \text{vec}(\boldsymbol{P'})}{\partial \text{vec}(\boldsymbol{Q'})} = (\boldsymbol{K'}^{\top} \otimes \boldsymbol{I}) \cdot \boldsymbol{C}_{dN},
$$

where $\boldsymbol{C}_{dN}$ is the communication matrix.

Then, we have

$$
\begin{aligned}
\boldsymbol{J}_Q^{Q'} &= \frac{\partial \text{vec}(\boldsymbol{Q'})}{\partial \text{vec}(\boldsymbol{Q})} = \text{blockdiag}\left( \frac{\partial \boldsymbol{q'}_1}{\partial \boldsymbol{q}_1}, \frac{\partial \boldsymbol{q'}_2}{\partial \boldsymbol{q}_2}, \dots, \frac{\partial \boldsymbol{q'}_N}{\partial \boldsymbol{q}_N} \right), \\
\boldsymbol{J}_K^{K'} &= \frac{\partial \text{vec}(\boldsymbol{K'})}{\partial \text{vec}(\boldsymbol{K})} = \text{blockdiag}\left( \frac{\partial \boldsymbol{k'}_1}{\partial \boldsymbol{k}_1}, \frac{\partial \boldsymbol{k'}_2}{\partial \boldsymbol{k}_2}, \dots, \frac{\partial \boldsymbol{k'}_N}{\partial \boldsymbol{k}_N} \right),
\end{aligned}
$$

where

$$
\frac{\partial \boldsymbol{q'}_i}{\partial \boldsymbol{q}_i} = \frac{\sqrt{d_h}}{\|\boldsymbol{q}_i\|} \text{diag}(\boldsymbol{\gamma}_q) \left( \boldsymbol{I} - \frac{\boldsymbol{q}_i \boldsymbol{q}_i^{\top}}{\|\boldsymbol{q}_i\|_2^2} \right).
$$

Moreover, we have

$$
\begin{aligned}
\frac{\partial \text{vec}(\boldsymbol{Q})}{\partial \text{vec}(\boldsymbol{X})} &= \boldsymbol{I} \otimes \boldsymbol{W}_q, \\
\frac{\partial \text{vec}(\boldsymbol{K})}{\partial \text{vec}(\boldsymbol{X})} &= \boldsymbol{I} \otimes \boldsymbol{W}_k.
\end{aligned}
$$

Thus, we have

$$
\frac{\partial \boldsymbol{q'}_i}{\partial \boldsymbol{x}_i} = \frac{\partial \boldsymbol{q'}_i}{\partial \boldsymbol{q}_i} \frac{\partial \boldsymbol{q}_i}{\partial \boldsymbol{x}_i} = \frac{\sqrt{d_h}}{\|\boldsymbol{q}_i\|_2} \text{diag}(\boldsymbol{\gamma}_q) \left( \boldsymbol{I} - \frac{\boldsymbol{q}_i \boldsymbol{q}_i^{\top}}{\|\boldsymbol{q}_i\|_2^2} \right) \boldsymbol{W}_q = \sqrt{d_h} \text{diag}(\boldsymbol{\gamma}_q) \left( \boldsymbol{I} - \frac{\boldsymbol{q}_i \boldsymbol{q}_i^{\top}}{\|\boldsymbol{q}_i\|_2^2} \right) \boxed{\frac{\boldsymbol{W}_q}{\|\boldsymbol{W}_q \boldsymbol{x}_i\|_2}},
$$

$$
\frac{\partial \boldsymbol{k'}_j}{\partial \boldsymbol{x}_j} = \frac{\partial \boldsymbol{k'}_j}{\partial \boldsymbol{k}_j} \frac{\partial \boldsymbol{k}_j}{\partial \boldsymbol{x}_j} = \frac{\sqrt{d_h}}{\|\boldsymbol{k}_j\|_2} \text{diag}(\boldsymbol{\gamma}_k) \left( \boldsymbol{I} - \frac{\boldsymbol{k}_j \boldsymbol{k}_j^{\top}}{\|\boldsymbol{k}_j\|_2^2} \right) \boldsymbol{W}_k = \sqrt{d_h} \text{diag}(\boldsymbol{\gamma}_k) \left( \boldsymbol{I} - \frac{\boldsymbol{k}_j \boldsymbol{k}_j^{\top}}{\|\boldsymbol{k}_j\|_2^2} \right) \boxed{\frac{\boldsymbol{W}_k}{\|\boldsymbol{W}_k \boldsymbol{x}_i\|_2}}.
$$

In Proposition 3, we have proved that in a high-dimensional setting, the singular values of $\frac{\boldsymbol{W}}{\|\boldsymbol{W}\boldsymbol{x}\|}$ is independent to the magnitude of $\boldsymbol{W}$. Thus, until now, we have proved the Proposition 5. $\qquad \square$

Further, we would like to discuss the Jacobian matrix $\frac{\partial \text{vec}(\boldsymbol{Y})}{\partial \text{vec}(\boldsymbol{X})}$ after QKNorm. We have that

$$
\frac{\partial \text{vec}(\boldsymbol{Y})}{\partial \text{vec}(\boldsymbol{X})} = (\boldsymbol{A'}^{\top} \otimes \boldsymbol{W}_v) + (\boldsymbol{I}_n \otimes \boldsymbol{W}_v \boldsymbol{X}) \frac{\boldsymbol{J}}{\sqrt{d_h}} \left( (\boldsymbol{K'}^{\top} \otimes \boldsymbol{I}) \cdot \boldsymbol{C}_{dN} \boldsymbol{J}_Q^{Q'} (\boldsymbol{I} \otimes \boldsymbol{W}_q) + (\boldsymbol{I} \otimes \boldsymbol{Q'}^{\top}) \boldsymbol{J}_K^{K'} (\boldsymbol{I} \otimes \boldsymbol{W}_k) \right).
\tag{17}
$$

It should be noted that:

- $K'$ and $Q'$ are two normalized terms that have relatively stable range of values.

- the Jacobian matrix of $J_Q^{Q'}$ is relatively independent to the magnitude of $Q$ and $J_K^{K'}$ is relatively independent to the magnitude of $K$ in a high-dimensional setting.

- QKNorm cannot fully replace the value of PreNorm because $\frac{\partial \text{vec}(Y)}{\partial \text{vec}(X)}$ in Equation 17 is directly affected by $X$.

We note that QKNorm will elliviate the influence of the magnitude of $W_q$ and $W_k$ on the Jacobian $\frac{\partial \text{vec}(Y)}{\partial \text{vec}(X)}$. In the traditional self-attention, $\frac{\partial \text{vec}(Y)}{\partial \text{vec}(X)}$ is largely affected by $W_q^\top W_k$. However, after QKNorm, $\frac{\partial \text{vec}(Y)}{\partial \text{vec}(X)}$ will only be affected by $W_q$ or $W_k$, independently, rather than their product (*i.e.*, $W_q^\top W_k$). This is important for the training stability because the singular values of $W_q^\top W_k$ will increase extremely fast when both singular values of $W_q$ and $W_k$ are increasing.

## C  EXPERIMENTAL DETAILS

TABLE 3: Model configurations, peak learning rate and weight decay for different optimizers.

| Acronym | Size | d_model | n_head | depth | AdamW | | mSGDW | |
|---|---|---|---|---|---|---|---|---|
| | | | | | LR | WD | LR | WD |
| L-DNT-Small | 124M | 768 | 12 | 12 | 6e-4 | 0.1 | 1.0 | 1e-4 |
| L-DNT-Large | 774M | 1280 | 20 | 36 | 6e-4 | 0.1 | 1.0 | 1e-4 |
| L-DNT-XL | 1436M | 1536 | 24 | 48 | 6e-4 | 0.1 | 1.0 | 1e-4 |
| V-DNT-Large | 307M | 1024 | 16 | 24 | 1e-3 | 0.1 | 0.5 | 2e-4 |
| V-DNT-Huge | 632M | 1280 | 16 | 32 | 1e-3 | 0.1 | 0.1 | 1e-3 |

We conduct experiments on two popular architectures: Vision Transformer (ViT) and Generative Pre-trained Transformer (GPT). Our implementation leverages established repositories: timm (Wightman, 2019) for ViT and nanoGPT (Karpathy, 2022) for GPT models. We utilized five model configurations: L-DNT-Small (124M parameters), L-DNT-Large (774M parameters), L-DNT-XL (1436M parameters), V-DNT-Large (307M parameters), and V-DNT-Huge (632M parameters). Model specifications including hidden dimension (d_model), number of attention heads (n_head), and network depth are detailed in Table 3. The training is conducted using PyTorch (Paszke et al., 2019) with bfloat16 precision GPUs, employing a cosine learning rate schedule.

All language models are trained on OpenWebText, using GPT-2 tokenizer. The training dataset contains 9B tokens, with a validation set of 4.4M tokens, following the train-validation split from nanoGPT. We employed distributed data parallel training with gradient accumulation. All models are trained using bfloat16 precision. The 124M models are trained on machines with 8 GPUs, 774M models are trained with 16 A800 GPUs, while 1436M models are trained with 32 GPUs. Our global batch sizes for 125M, 770M and 1436M models are 480, 512 and 512 individually. In Sophia (Liu et al., 2023b), they use 480 global batch size for all models. For all language models, we used 2000 steps or learning rate warmup to the maximum learning rate, and then used a cosine learning rate decay. It takes around four days to train 200K steps for the 1.4B model on 32 GPUs.

All vision models are trained on ImageNet dataset. We trained each 150 epoches as (Xie et al., 2024). We used a learning rate warmup of 60 epochs to the maximum learning rate, and then used a cosine learning rate decay.

For our experiments, we focus on comparing AdamW and mSGDW optimizers. The hyperparameters for AdamW are carefully tuned, with $\beta_1 = 0.9$ and $\beta_2 = 0.95$, following the dominant configuration in LLM pre-training literature. For weight decay, we used 0.1 for AdamW as (Karpathy, 2022; Liu

et al., 2023b). We used the recommended learning rate by nanoGPT for AdamW in GPT. Since our DNT is robust to large learning rate, we use $6 \times 10^{-4}$ for all our L-DNT models and $1 \times 10^{-3}$ for all our V-DNT models for AdamW. For mSGDW, we simply use a rough grid search for the learning rate, we cannot search a fine-grained learning rate and weight decay due to shortage in computation resources. For the momentum in mSGDW, we used a default 0.9 for all experiments. We use the implementation of mSGDW from timm (Wightman, 2019). This implementation is a decoupled weight decay regularization used in AdamW (Loshchilov & Hutter, 2019). Note that mSGDW is not directly to add a weight decay in the original implementation of mSGD in the official PyTorch, it will have performance problem.

TABLE 4: Training configurations for ViT and V-DNT.

| training config | ViT-L/H ($224^2$) | ViT-L/H ($224^2$) | V-DNT-L/H ($224^2$) | V-DNT-L/H ($224^2$) |
|---|---|---|---|---|
| optimizer | AdamW | mSGDW | AdamW | mSGDW |
| learning rate schedule | | cosine decay | | |
| peak learning rate | 1e-3 | 0.5/0.1 | 1e-3 | 0.5/0.1 |
| minimum learning rate | 1e-8 | 1e-8 | 1e-8 | 1e-8 |
| weight decay | 0.1 | 2e-4/1e-3 | 0.1 | 2e-4/1e-3 |
| optimizer momentum | $\beta_1, \beta_2 = 0.9, 0.99$ | $\mu = 0.9$ | $\beta_1, \beta_2 = 0.9, 0.99$ | $\mu = 0.9$ |
| warmup epoches | 60 | 60 | 60 | 60 |
| weight init | | Truncated Xavier | | |
| batch size | | 1024 | | |
| training epochs | | 150 | | |
| randaugment | | $(9, 0.5)$ | | |
| mixup | | 0.8 | | |
| cutmix | | 1.0 | | |
| random erasing | | 0 | | |
| label smoothing | | 0.1 | | |
| stochastic depth | | 0.1/0.5 | | |
| gradient clip | | None | | |
| exp. mov. avg. (EMA) | | no | | |

https://github.com/huggingface/pytorch-image-models/blob/main/timm/optim/sgdw.py
https://pytorch.org/docs/stable/generated/torch.optim.SGD.html

TABLE 5: Training configurations for GPT and L-DNT.

| training config | GPT2-S/L/XL | GPT2-S/L/XL | L-DNT-S/L/XL | L-DNT-S/L/XL |
|---|---|---|---|---|
| optimizer | AdamW | mSGDW | AdamW | mSGDW |
| learning rate schedule | | cosine decay | | |
| peak learning rate | 6e-4/2.5e-4/1.5e-4 | 1.0 | 6e-4 | 1.0 |
| minimum learning rate | 6e-5 | 6e-5 | 6e-5 | 6e-5 |
| weight decay | 0.1 | 1e-4 | 0.1 | 1e-4 |
| optimizer momentum | $\beta_1, \beta_2 = 0.9, 0.95$ | $\mu = 0.9$ | $\beta_1, \beta_2 = 0.9, 0.95$ | $\mu = 0.9$ |
| warmup steps | 2000 | 0 | 2000 | 0 |
| weight init | | Xavier | | |
| tokens seen each update | | 480K/512K/512K | | |
| max iters | | 200K | | |
| batch size | | 480/512/512 | | |
| sequence length | | 1024 | | |
| dropout | | 0.0 | | |
| bfloat16 | | True | | |
| gradient clipping | | 1.0 | | |

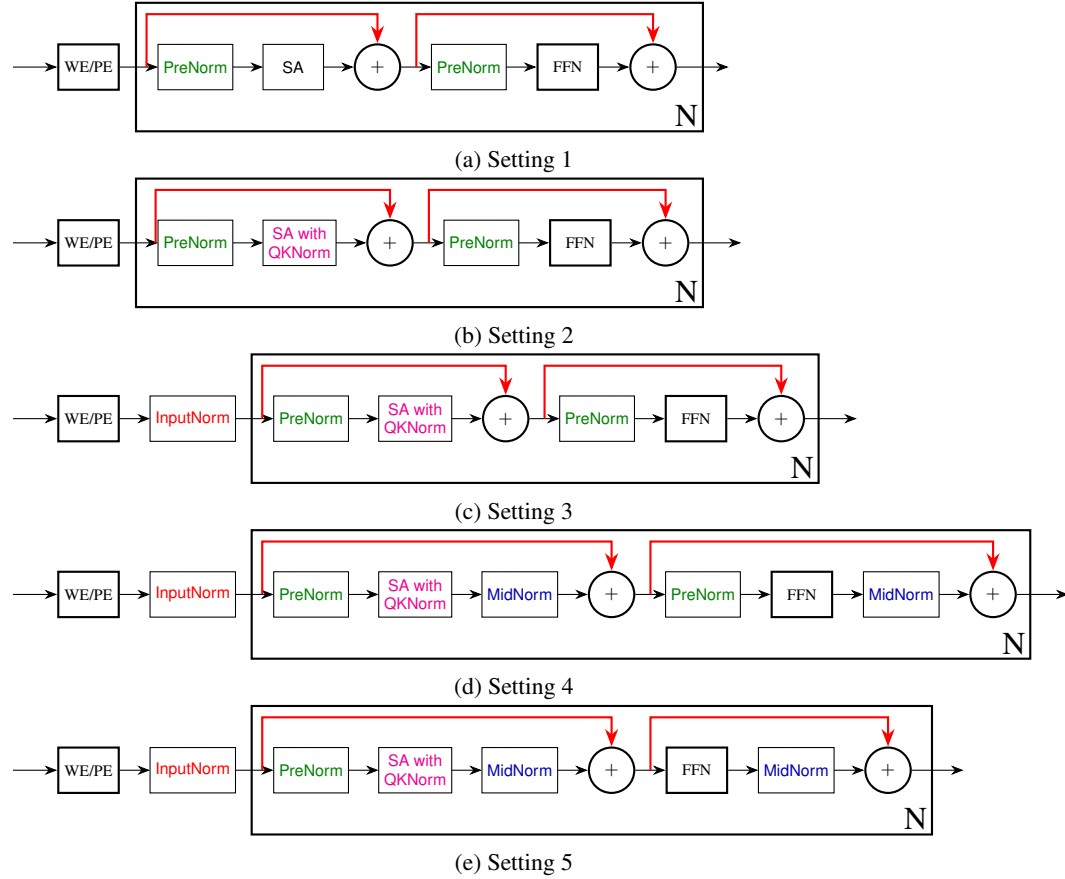

(a) Setting 1

(b) Setting 2

(c) Setting 3

(d) Setting 4

(e) Setting 5

FIGURE 7: Five different settings of normalizations evaluated in the ablation study. (a): the standard Transformer with PreNorm. (b): (a) + QKNorm. (c): (b) + InputNorm. (d) and (e): two versions of our DNTs.

# D FIVE DIFFERENT NETWORK SETTINGS

In Figure 7, we illustrate five different network settings and we have conducted a set of ablation studies for the five settings in the main body part.

# E EXPERIMENTS ON LARGER MODEL

We further compare larger V-DNT and L-DNT models and original ViT and GPT2 models on ImageNet and OpenWebText using mSGDW and AdamW. The results are shown in Figures 8, 9 and 10.

We see that on OpenWebText, L-DNT-large with mSGDW achieves a comparable performance with L-DNT-large with AdamW and achieves a much better performance than GPT2-large with mSGDW. Meanwhile, we find out that V-DNT-huge with mSGDW achieves a comparable performance with V-DNT-huge with AdamW and obtains a much better performance than ViT-huge with mSGDW.

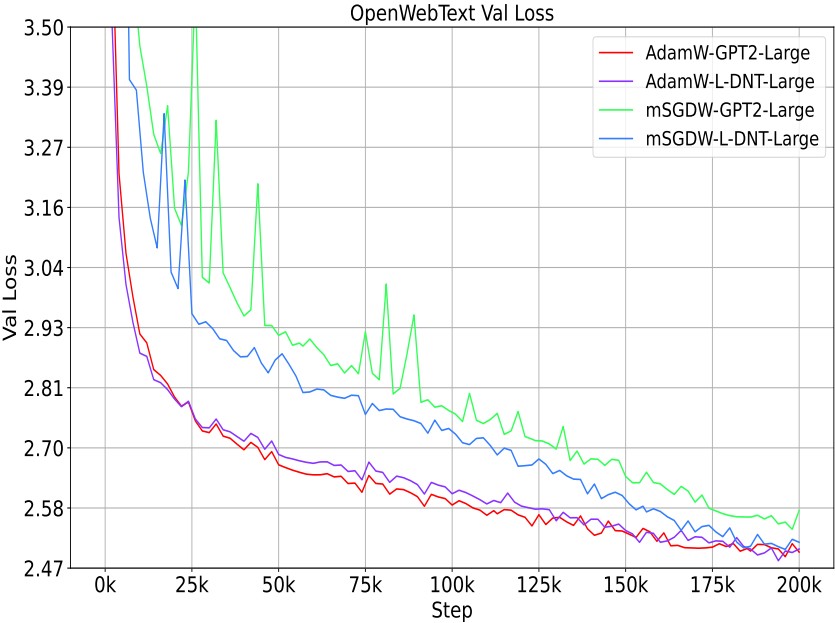

FIGURE 8: Comparison of GPT2-Large and L-DNT-Large (774M) on OpenWebText. All models are trained with 200K in total. GPT2-Large training mSGDW under-performs GPT2-Large with AdamW significantly, but L-DNT-Large with mSGDW can achieve a comparable performane with L-DNT-Large with AdamW.

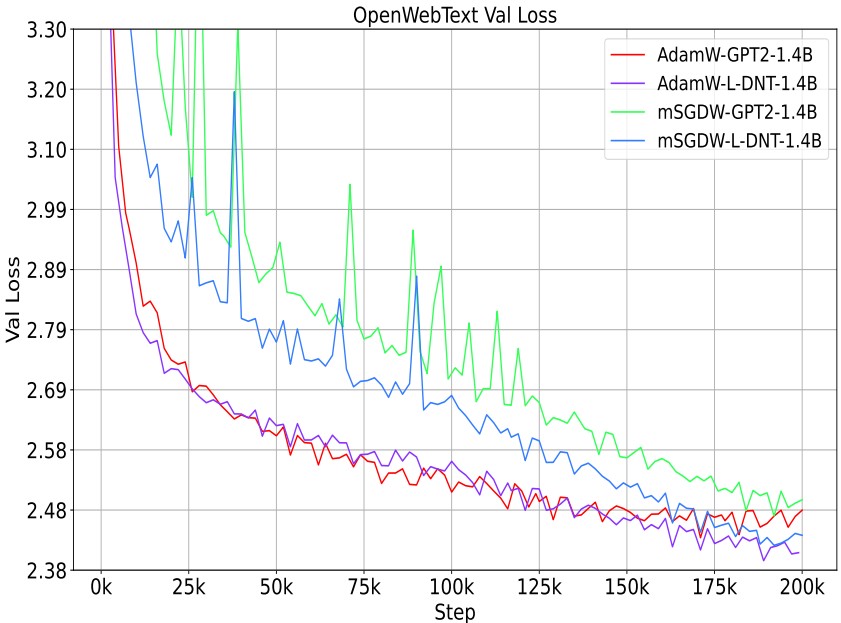

FIGURE 9: Comparison of GPT2-Large and L-DNT-Large (1436M) on OpenWebText. All models are trained with 200K in total. GPT2-Large training mSGDW under-performs GPT2-Large with AdamW significantly, but L-DNT-Large with mSGDW can achieve a comparable performane with L-DNT-Large with AdamW.

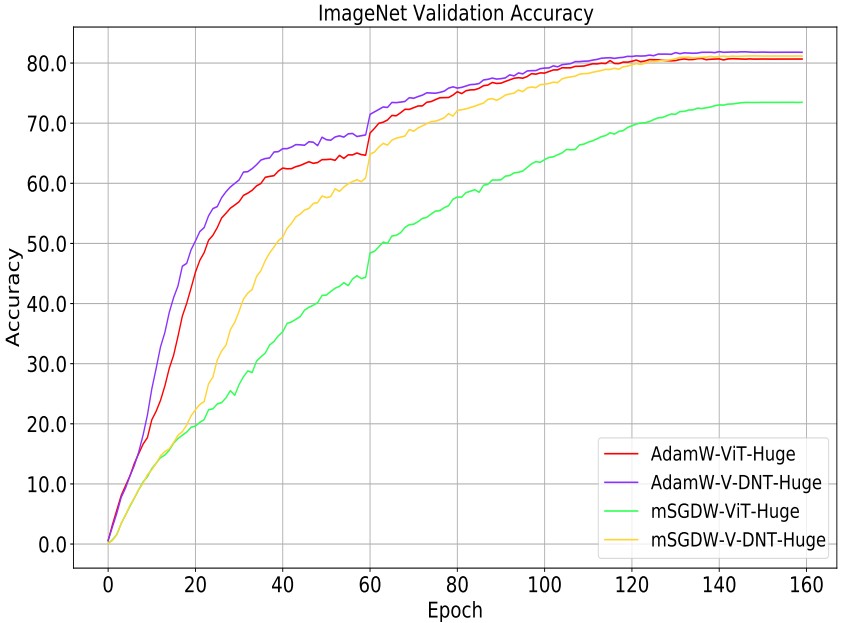

FIGURE 10: Comparison of ViT-Huge and V-DNT-Huge (632M) on ImageNet. All models are trained with 160 epochs in total. ViT-Huge training mSGDW under-performs ViT-Huge with AdamW significantly, but V-DNT-Huge with mSGDW can achieve a comparable performane with V-DNT-Huge with AdamW.

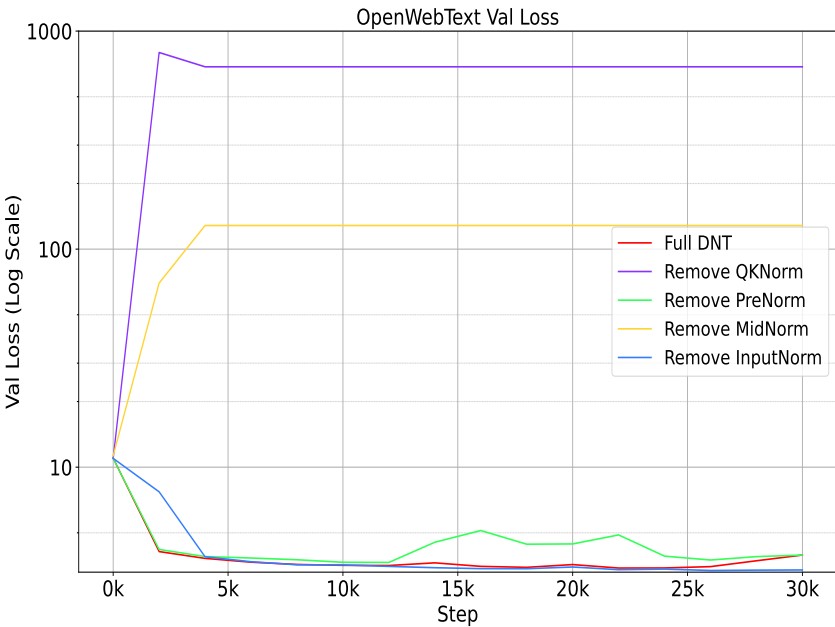

FIGURE 11: Ablation study on the influence of different normalization methods on training stability.

## F MORE ABLATION STUDY

In this section, we conducted more ablation studies. In these experiments, we started with a complete L-DNT model (we use setting 5 in Figure 7) that includes one InputNorm, two PreNorm layers, two MidNorm layers, and one QKNorm. We removed InputNorm, PreNorm (both layers), MidNorm (both layers), and QKNorm individually. To observe the instability issue, we trained each ablated model with a relatively large learning rate (Adam optimizer with learning rate 0.3, without warmup).

According to the experimental results, we observed that:

- When only removing QKNorm, the model collapsed after just a few training steps;

- When only removing PreNorm, the model began to collapse around 4000 steps;

- When only removing MidNorm, the model exhibited significant oscillations between 10K and 30K steps;

- When only removing InputNorm, the model was able to converge normally even with the 0.3 learning rate.

Therefore, from the perspective of training stability, we conclude that the importance of these normalization layers can be ranked in the following order: QKNorm > PreNorm ≈ MidNorm > InputNorm.

Meanwhile, we also conducted an ablation study on the V-DNT model using AdamW, and we observed a similar phenomenon. QKNorm is crucial, while PreNorm and MidNorm exhibit comparable stability. InputNorm has a relatively minor impact on stability.

### F.1 EXPERIMENTS USING MUON OPTIMIZER

In this section, we conducted experiments with using the Muon optimizer to compare GPT-2 and DNT. First of all, we give a brief introduction to Muon: Given an update $G$ and its SVD decomposition, say $G = U\Sigma V^\top$. Muon uses a Newton-Schulz method to approximate the $UV^\top$ matrix. In the current Muon implementation, for matrices, we use Muon to approximate the matrices, while for vectors, we directly employ the AdamW optimizer.

In our experiments with Muon, we use a learning rate of $10^{-4}$ for matrices and $6 \times 10^{-4}$ for vectors. We conducted four sets of experiments: a) GPT-2 with AdamW, b) DNT with AdamW, c) GPT-2 with Muon, and d) DNT with Muon. Same as previous experiments, we use 2000 steps warmup for all four experiments. The results are presented in Figure 12.

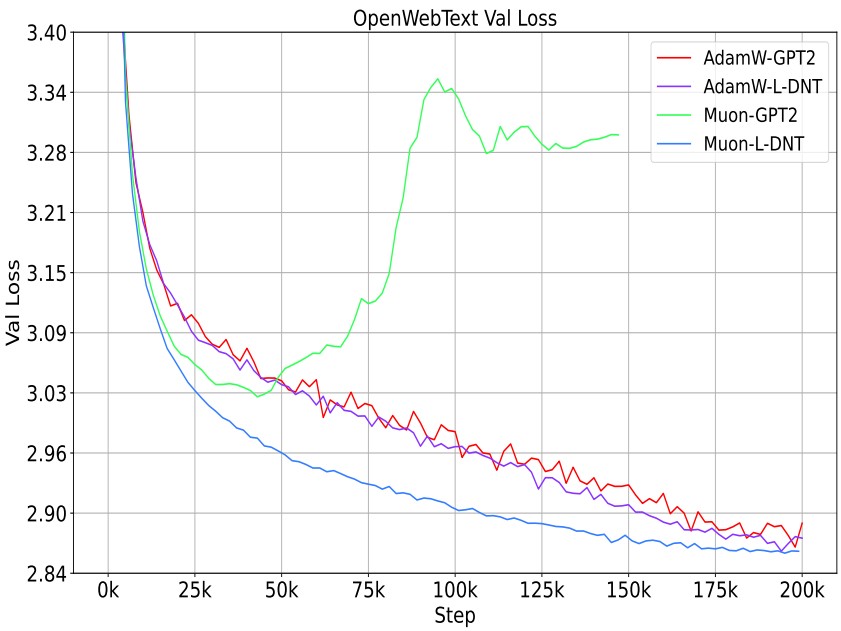

FIGURE 12: Performance comparison of DNT and GPT using different optimizers.

Based on the observations in experiments, we note the following two points.

- When using the learning rates $(10^{-4}, 6 \times 10^{-4})$, GPT-2 begins to show an increase in validation loss after 50K training steps, indicating training instability. For DNT, we did not observe any instability issues, and the training curve exhibits almost no fluctuations. Additionally, when using Muon, our DNT demonstrates significantly better convergence speed compared to DNT with AdamW.

- When both DNT and GPT-2 use AdamW, our DNT shows slightly better performance than GPT-2.

## G    A BRIEF INTRODUCTION TO SOME EXISTING OPTIMIZERS

Optimization methods in deep learning can be broadly categorized into first-order methods and second-order methods, each with distinct characteristics and applications. First-order optimization algorithms dominate deep learning due to their computational efficiency, particularly for high-dimensional and large-scale problems. First-order methods rely primarily on gradient information to find the minimum or maximum of a function. Based on learning rate selection strategies. These methods can be divided into optimizers with fixed step size and optimizers with adaptive learning rate.

Stochastic Gradient Descent (SGD) (Robbins & Monro, 1951) serves as the foundational algorithm for neural network optimization. It updates parameters in the opposite direction of the gradient of the objective function. While simple and effective, vanilla SGD can struggle with navigating ravines and saddle points in the loss landscape. Momentum SGD (mSGD) (Nesterov, 1983) addresses the limitations of vanilla SGD by accelerating gradient descent in relevant directions while dampening oscillations. This method augments the gradient direction with a fraction of the update vector from the previous step, allowing faster convergence and helping escape local minima. Other notable variants include signSGD (Bernstein et al., 2018), which uses only the sign of gradients for updates;

SVRG (Johnson & Zhang, 2013), which reduces variance in stochastic gradients; LARS (You et al., 2017), which adjusts learning rates layer-wise.

Adaptive methods revolutionized gradient-based optimization by incorporating two key innovations. First, they implement parameter-specific learning rate adaptation, performing smaller updates for frequently occurring features and larger updates for less frequent features. Second, they incorporate historical gradient information, often approximating second-order properties of the loss landscape. AdaGrad (Duchi et al., 2011) adapts learning rates based on historical gradient information and is particularly effective for sparse data. RMSprop (Hinton, 2012) addresses AdaGrad's radically diminishing learning rates by using an exponentially weighted moving average. Adam (Kingma & Ba, 2014) combines momentum with adaptive learning rates, incorporating both first and second moments of gradients. AdamW (Loshchilov & Hutter, 2019) modifies Adam with more effective weight decay regularization, while Adafactor (Shazeer & Stern, 2018) provides a memory-efficient adaptive method. Défossez et al. provides a unified formulation for adaptive methods like AdaGrad, Adam, and AdaDelta.

The optimization field continues to evolve with recent innovations including MUON (Jordan et al., 2024), LION (Chen et al., 2023), Sophia (Liu et al., 2023b), and Mars (Yuan et al., 2024). These methods represent the cutting edge of adaptive optimization techniques, further advancing efficiency and performance in training deep learning models.

Wang & Choromanska (2025) give a detailed analysis and survey of optimization methods, we would like to recommend the reader to read it for a full reference.

**Remark.** This paper is orthogonal to these works discussed in this section. Our primary contribution is to demonstrate that vanilla mSGD can achieve strong performance on a Transformer architecture when it does not have a heavy-tail problem in gradients. Notably, the optimizers discussed here can also be effectively applied to our proposed DNT network.

