# OpenReview forum: "DNT: a Deeply Normalized Transformer that can be trained by Momentum SGD"
_ICLR.cc/2026/Conference — ICLR 2026 Poster_

### Official Review · Reviewer_gReu · 2025-10-29

**Soundness:** 3
**Presentation:** 3
**Contribution:** 3
**Rating:** 4
**Confidence:** 4

**Summary:**

This manuscript targets the demand for advanced optimizers with adaptive learning rates in the training of Transformers. A deep normalization transformer is proposed, in which the heavy-tail gradient problem is overcome by strategically integrating different normalization techniques into the appropriate positions of the Transformer.

**Strengths:**

The idea is timely, clearly presented, theoretically analyzed, and empirically evaluated on two popular Transformer architectures.

**Weaknesses:**

However, key clarifications and stronger validation are necessary. The detailed comments are as follows:

1. Based on the normalization techniques mentioned in Figure 4, is the contribution of this manuscript merely an engineering improvement regarding the strategic integration of different normalization techniques? What is the logic of cooperation between them? And how to determine the appropriate positions of each technology in the Transformer?

2. In Tables 2-4 of Appendix C, is the selection of parameters specified by the authors or the relatively optimal settings obtained through search? In the experiment, key hyperparameters and design choices were lacking in ablation.

3. There are inconsistencies or errors in the reference writing, such as:
"Xiangning Chen, Chen Liang, Da Huang, Esteban Real, Kaiyuan Wang, Hieu Pham, Xuanyi Dong, Thang Luong, Cho-Jui Hsieh, Yifeng Lu, et al. Symbolic discovery of optimization algorithms. Advances in neural information processing systems, 36, 2024."
should be:
"Xiangning Chen, Chen Liang, Da Huang, Esteban Real, Kaiyuan Wang, Hieu Pham, Xuanyi Dong, Thang Luong, Cho-Jui Hsieh, Yifeng Lu, et al. Symbolic discovery of optimization algorithms. Advances in neural information processing systems, 36:49205-49233, 2023."
It is suggested that authors should carefully check, revise and improve.

4. In line 426, "See Appendix H for the training parameters." In fact, "Appendix H" does not exist in the manuscript.

5. In the manuscript, "the norm of * is very large", "the norm of * is large" and "* become too large" are mentioned times. How to define or quantitatively describe "very large", "large", or "too large"?

6. What is the definition of σ(∙) in lines 303-304 σ(W_1 ) and σ(W_2 )? In Equation (4), "Y = Self-Attention(X′), where x′ = PreNorm(x) ". However, what kind of variable is X′? Or is there any connection between X' and x'? Furthermore, in line 256, "Y = Self-Attention(X) and Y' = Self-Attention(X')". Is it accurate that it is different from the "Y = Self-Attention(X')" in equation (4)?

7. In order to facilitate readers' understanding, the authors should introduce and explain the alphabetic symbols and various operation symbols that appear in the manuscript. For example: The definition of "⨂" in the equation on lines 241-242 and in Equation (5) needs to be declared. What is "C" in Equation (5)? The "⨀" in line 254 needs to declare its definition. Furthermore, x_0, which appears in lines 372-373, seems not to have been used. So, where does it play any role? What parameter is "C_dn" in the formula on lines 730-731? What is "C" in Equation (13)?

8. There is a cross-reference error in "according to Equation ??" in line 754. Please check and confirm whether other citations are standardized.

9. In lines 759-758, there are ∂L/(∂vec(Y)) and  ∂L/(∂vec(W_q)),   ∂L/(∂vec(W_k)),   ∂L/(∂vec(W_v)), where L and L should be the same?

10. The captions in Figures 7-10 are not consistent with those of the other figures mentioned earlier in the manuscript.

11. What are the dimensions W_1 and W_2 that appear in Equation (6) respectively? In line 274, does the W_1 x>0 in "W_2 diag(1(W_1 x>0)) W_1" represent the relationship between the vector W_1 x and 0? How exactly is it defined? Furthermore, what kind of calculation is 1(W_1 x>0)? The authors should supplement the corresponding definitions or calculation methods.

12. The "if" in line 754 should be "If".

**Questions:**

see above section for detailed information

---

> ### Author Response · Authors · 2025-11-22
> **Author responses to Reviewer gReu (Part 1)**
>
> We would like to sincerely thank the reviewer for your careful reading of our paper and your constructive suggestions.
>
> To address your concerns, we have added further clarified our key contributions, more experiments, and revised the paper according to your suggestions. We really hope you will be satisfied with our responses.
>
> **Q1: Reviewer's comments on the contribution of this manuscript**
>
> > Based on the normalization techniques mentioned in Figure 4, is the contribution of this manuscript merely an engineering improvement regarding the strategic integration of different normalization techniques? What is the logic of cooperation between them? And how to determine the appropriate positions of each technology in the Transformer?
>
> **Our response**: Previous works have pointed out that a heavy-tailed distribution of the stochastic gradients is a root cause of SGD's poor performance. We observe that
> the heavy-tail problem in gradients is indeed closely related to the large diversity of the singular values in the Jacobian matrix $\frac{\partial {\boldsymbol{x}^{l+1}}}{\partial \boldsymbol{x}^{l}}$. The Jacobian matrix can have highly diverse singular values for several reasons: 1) the weight matrix contains very diverse singular values; 2) the activations span widely, leading to Jacobians with very uneven singular value distributions. When a matrix has a wide range of singular values (\ie, a very large condition number), it means that the transformation stretches the input
> very differently along different directions. During backpropagation, it will cause a heavy-tail problem in the gradients.
>
> *Therefore, a golden rule for deciding whether to add a normalization layer is to determine if the block is likely to produce a Jacobian matrix $\frac{\partial {\boldsymbol{x}^{l+1}}}{\partial \boldsymbol{x}^{l}}$ with a highly non-uniform distribution of singular values. If so, we need to incorporate a normalization layer. Determining its placement also involves assessing whether, after adding the normalization layer, the distribution of singular values in the Jacobian matrix is more likely to be uniform and free of particularly abnormal values.*
>
> In the revised paper, we supply more ablation studies. In these experiments, we started with a complete DNT model that includes one InputNorm, two PreNorm layers, two MidNorm layers, and one QKNorm. We removed InputNorm, PreNorm (both layers), MidNorm (both layers), and QKNorm individually. We trained each ablated model with a relatively large learning rate (Adam optimizer with learning rate 0.3, without warmup) to observe when the training would collapse.
>
> We observed that:
>
> - When only removing QKNorm, the model collapsed after just a few training steps;
> - When only removing PreNorm, the model began to collapse around 4000 steps;
> - When only removing MidNorm, the model exhibited significant oscillations between 15K and 25K steps;
> - When only removing InputNorm, the model was able to converge normally even with the 0.3 learning rate
>
> Therefore, from the perspective of training stability, we conclude that the importance of these normalization layers can be ranked in the following order: QKNorm > PreNorm $\approx$ MidNorm > InputNorm.
>
> Meanwhile, we also had an ablation study that started from the GPT baseline and then sequencially added each norm.
>
> **Q2: On the parameter in Table2-4 of Appendix C**
>
> > In Tables 2-4 of Appendix C, is the selection of parameters specified by the authors or the relatively optimal settings obtained through search? In the experiment, key hyperparameters and design choices were lacking in ablation.
>
> **Our response:** Thank you very much for your comment. We would clarify the selection of parameters in Appendix C. For AdamW in ViT, we follow the standard Timm settings. For AdamW in GPT, we followed the settings in GPT3 and Sophia, we slightly tune AdamW around the suggested values and reported a better results. For our DNT, it is expensive to search the parameter in a large scale. For AdamW in V-DNT, we use the similar parameters as in ViT and GPT, but since our DNT can tolerent a larger learning rate, we use a larger learning rate, but we do not tune it to the best. You can see that, for L-DNT-Small, L-DNT-Large, and L-DNT-XL, we use 6e-4 for all three models. Be honest, it is hard for us to tune the parameter in fine scale.

---

> ### Author Response · Authors · 2025-11-22
> **Author responses to Reviewer gReu (Part 2)**
>
> **Q3: On inconsistencies in the reference writing**
>
> > There are inconsistencies or errors in the reference writing, such as: "Xiangning Chen, Chen Liang, Da Huang, Esteban Real, Kaiyuan Wang, Hieu Pham, Xuanyi Dong, Thang Luong, Cho-Jui Hsieh, Yifeng Lu, et al. Symbolic discovery of optimization algorithms. Advances in neural information processing systems, 36, 2024." should be: "Xiangning Chen, Chen Liang, Da Huang, Esteban Real, Kaiyuan Wang, Hieu Pham, Xuanyi Dong, Thang Luong, Cho-Jui Hsieh, Yifeng Lu, et al. Symbolic discovery of optimization algorithms. Advances in neural information processing systems, 36:49205-49233, 2023." It is suggested that authors should carefully check, revise and improve.
>
> **Our response:** Thank you for pointing out this issue. We have fixed the error and checked the format of the other references.
>
> **Q4: On Appendix H**
>
> > In line 426, "See Appendix H for the training parameters." In fact, "Appendix H" does not exist in the manuscript.
>
> **Our response:** Sorry for the typo. We previously split the training parameters into two sections and later merged them, but failed to update the corresponding reference. We have now fixed this issue. Thank you for your careful reading.
>
> **Q5: On how to evaluate the norm is large or not?**
>
> > In the manuscript, "the norm of * is very large", "the norm of * is large" and "* become too large" are mentioned times. How to define or quantitatively describe "very large", "large", or "too large"?
>
> **Our response:** We are sorry for the confusion. When we say "the norm is very large," it usually indicates that such a large value may cause the network training to become unstable or crash. Given a vector $\boldsymbol{x} \in \mathcal{R}^d$, according to our observations, if the norm of $\boldsymbol{x}$ exceeds $10^5$ (sometimes the network collapses, we can see the norm is around $10^7$.), the network is likely to enter an unstable (or crashing) training state.
>
> **Q6: On definitions of some notations**
>
> > What is the definition of σ(∙) in lines 303-304 σ(W_1 ) and σ(W_2 )? In Equation (4), "Y = Self-Attention(X′), where x′ = PreNorm(x) ". However, what kind of variable is X′? Or is there any connection between X' and x'? Furthermore, in line 256, "Y = Self-Attention(X) and Y' = Self-Attention(X')". Is it accurate that it is different from the "Y = Self-Attention(X')" in equation (4)?
>
> **Our response:** When we mention $\sigma(\boldsymbol{W_1})$, we refer to the largest singular value of $\boldsymbol{W}_1$. We should denote it as $\sigma_1(\boldsymbol{W_1})$ and clarify that this represents the largest singular value of $\boldsymbol{W}_1$. We have already made this correction in the paper.
>
> In "Y = Self-Attention(X′), where X′ = PreNorm(X)", we mean that we first normalize each row of X and obtained X', then perform self-attention on the normalized X′. Y = Self-Attention(X) means that we directly apply a self-attention on X.
>
> In Proposition 2, we mean that if only a fixed scale vector is applied along the dimension of X, the normalization layer can cancel out the effect of this scaling after normalization. In this way, the normalization layer can prevent large norms of x from causing unstable effects on self-attention.
>
> **Q7: On some notations**
>
> > In order to facilitate readers' understanding, the authors should introduce and explain the alphabetic symbols and various operation symbols that appear in the manuscript. For example: The definition of "⨂" in the equation on lines 241-242 and in Equation (5) needs to be declared. What is "C" in Equation (5)? The "⨀" in line 254 needs to declare its definition. Furthermore, x_0, which appears in lines 372-373, seems not to have been used. So, where does it play any role? What parameter is "C_dn" in the formula on lines 730-731? What is "C" in Equation (13)?
>
> **Our response:** Thank you very much for your suggestions. In the revised paper, we have added more explanations for the meanings of some symbols:
>
> - The "⨂" in Equation (5) denotes the Kronecker product.
> - The "C" in Equation (5) represents the commutation matrix.
> - The "⨀" in line 254 indicates element-wise multiplication.
> - $\boldsymbol{x}^0$ denotes the input to the 0-th layer.
> - $\boldsymbol{C}_{dn}$ represents a commutation matrix that maps from a $dn \times dn$ matrix to another $dn \times dn$ matrix.
>
> Let me explain what a commutation matrix is. Suppose we have a matrix
>
> $$
> \boldsymbol{A} = \begin{bmatrix}
> 1 & 2 \\\\
> 3 & 4
> \end{bmatrix}
> $$
>
> and we want to transform it into another matrix $\boldsymbol{B}$, where $\boldsymbol{B} = \begin{bmatrix}
> 4 & 3 \\\\
> 2 & 1
> \end{bmatrix}$.
>
> In this case, the commutation matrix $\boldsymbol{C} = \begin{bmatrix}
> 0 & 0 & 0 & 1\\\\
> 0 & 0 & 1 & 0\\\\
> 0 & 1 & 0 & 0\\\\
> 1 & 0 & 0 & 0
> \end{bmatrix}$.
>
> We have $\text{vec}(\boldsymbol{B}) = \boldsymbol{C} \cdot \text{vec}(\boldsymbol{A})$.
>
> The "C" in Equation (13) is the same as in Equation (5) - it denotes the commutation matrix.

---

> > ### Author Response · Authors · 2025-11-22
> > **Author responses to Reviewer gReu (Part 3)**
> >
> > **Q8: On a typo of reference**
> >
> > > There is a cross-reference error in "according to Equation ??" in line 754. Please check and confirm whether other citations are standardized.
> >
> > **Our response:** Thank you for pointing out this typo. In the revised paper, we have carefully verified the references and made corresponding modifications to the manuscript.
> >
> > **Q9: On one notation L**
> >
> > > In lines 759-758, there are ∂L/(∂vec(Y)) and ∂L/(∂vec(W_q)), ∂L/(∂vec(W_k)), ∂L/(∂vec(W_v)), where L and L should be the same?
> >
> > __Our response:__ When typesetting the mathematical notation, we used the wrong font. The symbol $L$ you mentioned should indeed be $\mathcal{L}$. Thank you for carefully reading our paper.
> >
> > **Q10: On captions in Figures 7-10**
> >
> > > The captions in Figures 7-10 are not consistent with those of the other figures mentioned earlier in the manuscript.
> >
> > **Our response:** Thank you for pointing out this issue. We have rewritten the captions for Figures 7-10.
> >
> > **Q11: On the dimensions W_1 and W_2**
> >
> > > What are the dimensions W_1 and W_2 that appear in Equation (6) respectively? In line 274, does the W_1 x>0 in "W_2 diag(1(W_1 x>0)) W_1" represent the relationship between the vector W_1 x and 0? How exactly is it defined? Furthermore, what kind of calculation is 1(W_1 x>0)? The authors should supplement the corresponding definitions or calculation methods.
> >
> > **Our response:**
> >
> > Suppose the input $\boldsymbol{x} \in \mathbb{R}^d$ in GPT-2, then $\boldsymbol{W}_1 \in \mathbb{R}^{4d \times d}$, and $\boldsymbol{W}_2 \in \mathbb{R}^{d \times 4d}$.
> >
> > In $\operatorname{diag} (\boldsymbol{1}({ \boldsymbol{W}_1 \boldsymbol{x} > \boldsymbol{0}}))$, $\boldsymbol{1}(\cdot)$ represents an indicator function. For example, if
> >
> > $\boldsymbol{W}_1 \boldsymbol{x} = \begin{bmatrix} -0.1 \\\\ 0.2 \end{bmatrix}$
> >
> > then $\boldsymbol{1}({\boldsymbol{W}_1 \boldsymbol{x} > \boldsymbol{0}}) = \begin{bmatrix} 0 \\\\ 1 \end{bmatrix}$
> >
> > $\operatorname{diag} (\boldsymbol{1}({\boldsymbol{W}_1 \boldsymbol{x} > \boldsymbol{0}})) = \begin{bmatrix} 0 & 0 \\\\ 0 & 1 \end{bmatrix}$
> >
> > We have provided the corresponding definitions in the revised paper.
> >
> > **Q12: On a typo**
> >
> > > The "if" in line 754 should be "If".
> >
> > **Our response:** Thank you for pointing out this typo. We fixed it.

---

> > > ### Author Response · Authors · 2025-11-25
> > > **Response to Reviewer gReu**
> > >
> > > We sincerely thank the reviewer for the time and effort in reviewing our paper. We have largely strengthened the paper according to your comments. We hope our clarifications and additional experiments address your concerns. Please let us know if any questions remain, and thank you again for your time and effort.
> > >
> > > Best regards
> > >
> > > Authors

---

### Official Review · Reviewer_epVN · 2025-10-30

**Soundness:** 3
**Presentation:** 3
**Contribution:** 2
**Rating:** 4
**Confidence:** 3

**Summary:**

This paper addresses a key limitation in training Transformers: the poor performance of momentum SGD (mSGD) compared to adaptive optimizers like AdamW. The root cause is identified as the heavy-tailed distribution of gradients in Transformers, which causes uneven updates across parameters.

To resolve this, the authors propose Deeply Normalized Transformers (DNT), which strategically apply normalization at specific positions in the architecture to modulate Jacobians, balance weight and activation contributions, and reduce the heavy-tail behavior of gradients. Theoretical justifications are provided for why these normalization choices improve training with mSGD. Empirically, DNT achieves performance comparable to AdamW on ImageNet (Vision Transformers) and OpenWebText (GPT) while requiring less memory and computation, thanks to the ability to use mSGDW.

**Strengths:**

1. The paper addresses a well-known problem in training Transformers, namely why momentum SGD (mSGD) tends to fail compared to adaptive optimizers like AdamW. It provides a detailed theoretical analysis showing how the placement of normalization layers affects the conditioning of Jacobian matrices and the variance of gradients, explaining why these adjustments are crucial for stable training.

2. The approach is practical, leveraging existing normalization techniques in new positions without introducing additional components. Empirically, the proposed Deeply Normalized Transformer (DNT) matches AdamW performance on ImageNet and OpenWebText, with gradients that are more concentrated and stable under mSGDW.

3. Using mSGDW also reduces memory and computational requirements compared to AdamW, which is a significant practical advantage.

4. Finally, the paper offers analytical insights linking the structure of normalization to optimizer behavior, providing a deeper understanding of the interplay between architecture and training dynamics.

**Weaknesses:**

1. The paper has several limitations regarding the scope and scale of its experiments. It evaluates the proposed Deeply Normalized Transformer (DNT) only on two benchmarks, ImageNet and OpenWebText, and does not include large-scale or multimodal tasks. This narrow evaluation makes it difficult to assess how well the method generalizes to other domains or to the training of state-of-the-art large models.

2. Another limitation is the increased complexity introduced by multiple normalization placements. While these placements are key to stabilizing mSGD, they also add implementation overhead and require careful hyperparameter tuning.

3. The paper also lacks comparisons with other recent approaches designed to improve stability in Transformers, such as nGPT, Stable Transformer, or LipsFormer. Similarly, there is no evaluation against newer optimizers like Muon, which could provide important context for the relative benefits of DNT and mSGDW (Muon is closer to SGD than Adam).

4. Finally, the evaluation on GPT2-small is somewhat limited in scale and may not reflect the challenges of training modern large language models. The optimizer shows instabilities in some experiments, and the loss gap between mSGDW and AdamW remains non-negligible, which could be a critical concern for training larger or more complex architectures.

**Questions:**

See weaknesses.

---

> ### Author Response · Authors · 2025-11-22
> **Author responses to Reviewer epVN (Part 1)**
>
> We would like to express our thanks to the reviewer for the constructive suggestions. In the revised paper, we have added more experiments, clarified our key contributions, and revised the paper according to your suggestions. Hope you will be satisfied with our revisions.
>
> **Q1: Reviewer's comments on scope and scale of its experiments**
>
> > The paper has several limitations regarding the scope and scale of its experiments. It evaluates the proposed Deeply Normalized Transformer (DNT) only on two benchmarks, ImageNet and OpenWebText, and does not include large-scale or multimodal tasks. This narrow evaluation makes it difficult to assess how well the method generalizes to other domains or to the training of state-of-the-art large models.
>
> **Our response:**
>
> Thank you very much for your suggestion. In the revised paper, we have emphasized our large-scale experiments in Appendix E. We conducted experiments using a 1.4B language model, which we refer to as L-DNT-1.4B. Based on the experimental results, L-DNT-1.4B demonstrates better performance compared to its counterpart, GPT-2.
>
> Regarding the model size and dataset used in the experiments, we would like to note that in a recent and well-regarded optimization evaluation study [1], the model sizes evaluated ranged from 130M to 1.2B, and they only compared results on language modeling task. ImageNet and OpenWebText are highly large-scale and representative datasets. In fact, training an L-DNT-1.4B model requires hundreds of GPU days. We hope the reviewers can take this into consideration.
>
> [1] Wen, Kaiyue, David Hall, Tengyu Ma, and Percy Liang. "Fantastic pretraining optimizers and where to find them." *arXiv preprint arXiv:2509.02046* (2025).
>
> **Q2: On computational complexity**
>
> > Another limitation is the increased complexity introduced by multiple normalization placements. While these placements are key to stabilizing mSGD, they also add implementation overhead and require careful hyperparameter tuning.
>
> **Our response:** Thank you very much for your comment. In our DNT, InputNorm is used only once, so it has very slight impact on computational speed. In our DNT architecture, we employ 5 or 6 normalization layers in total. Compared to the standard GPT-2 model, we use 3 additional normalization layers, resulting in a computational overhead which is 6% higher than GPT-2. When compared to a GPT-2 model that uses QKNorm (which employs 4 normalization layers), our computational overhead is only 2% higher. Overall, the additional computational overhead is relatively small.

---

> ### Author Response · Authors · 2025-11-22
> **Author responses to Reviewer epVN (Part 2)**
>
> **Q3: On comparison with nGPT, StableTransformer, and Lipsformer**
>
> > The paper also lacks comparisons with other recent approaches designed to improve stability in Transformers, such as nGPT, Stable Transformer, or LipsFormer. Similarly, there is no evaluation against newer optimizers like Muon, which could provide important context for the relative benefits of DNT and mSGDW (Muon is closer to SGD than Adam).
>
> **Our response:** Thank you for your suggestion. We want to clarify that, in this paper, *we analyze the effects of normalization in different positions from a theoretical perspective*. Specifically, we explain:
>
> - Why we need QKNorm for self-attention? QKNorm can relieve or even remove the influence of the magnitude of $\boldsymbol{W}_q$ and $\boldsymbol{W}_k$ on the Jacobian matrix of self-attention, and thus reduce the risk of model  crash.
>
> - Why we need PreNorm for self-attention? Because PreNorm can constrain the norm of each column in activations X in each timestep, and thus amend the Jacobian matrix of self-attention to not be significantly affected by the magnitude of $\boldsymbol{X}$.
>
> - Why we need MidNorm? MidNorm will amend the Jacobian matrix of each sub-block (i.e., the sub-block with self-attention and the sub-block with FFN) in our DNT to not be affected by the magnitude of $\boldsymbol{W}_1$, $\boldsymbol{W}_2$, $\boldsymbol{W}_o$,$\boldsymbol{W}_v$.
>
> It should be noted that none of nGPT, StableTransformer, and LipsFormer explains where should we put a norm or not?
>
> We agree with your comment that ``Muon is closer to SGD than Adam''. mSGDW is obtained directly from the gradient with a momentum, while AdamW is derived by dividing the first moment by the square root of the second moment.
>
> We conducted experiments using the optimizer Muon to compare GPT-2 and DNT. Given an update $\boldsymbol{G}$, performing SVD decomposition, we have $\boldsymbol{G} = \boldsymbol{U} \boldsymbol{\Sigma} \boldsymbol{V}^{\top}$. As we know, Muon uses a Newton-Schulz method to approximate the $\boldsymbol{U} \boldsymbol{V}^{\top}$ matrix. In the current Muon implementation, for matrices, we use Muon to approximate the matrices, whereas for vectors, we directly employ the optimizer AdamW.}
>
> In our experiments with Muon, we used a learning rate of $10^{-4}$ for matrices and $6\times 10^{-4}$ for vectors. We conducted four sets of experiments: GPT-2 with AdamW, DNT with AdamW, GPT-2 with Muon, and DNT with Muon. Same as previous experiments, we use 2000 steps warmup for all four experiments.
>
> Based on experimental observations, we note the following two points:
> - When using the learning rates ($10^{-4}$, $6\times 10^{-4}$), GPT-2 begins to show an increase in validation loss after 50K training steps, indicating training instability. For DNT, we observe no instability issues, and the training curve exhibits almost no fluctuations. Additionally, when using Muon, DNT demonstrates significantly better convergence speed compared to DNT with AdamW.
> - When both DNT and GPT-2 use AdamW, DNT shows slightly better performance than GPT-2.
>
> **Q4: On the evaluation**
>
> > Finally, the evaluation on GPT2-small is somewhat limited in scale and may not reflect the challenges of training modern large language models. The optimizer shows instabilities in some experiments, and the loss gap between mSGDW and AdamW remains non-negligible, which could be a critical concern for training larger or more complex architectures.
>
> **Our response:** Thank you very much for your comments and your suggestions. We would like to claim that our core contribution in this paper is that ``we introduce a Deeply Normalized Transformer (DNT), that is meticulously
> engineered to overcome the heavy-tailed gradients issue''. In this paper, to show that strong stability of the DNT, we verify that DNT can be well trained with mSGDW. We agree with your comment that "the loss gap between mSGDW and AdamW remains non-negligible". In this paper, we demonstrate that DNT achieves a better performance than GPT2 using the same optimizer, and DNT shows a better stability than GPT.
>
> *Regarding the scale of evaluated models*, we conducted experiments using a 1.4B language model, which we refer to as L-DNT-1.4B. Based on the experimental results, L-DNT-1.4B demonstrates better performance compared to its counterpart, GPT-2.
>
> We also ran experiments on a 7B model for 10K steps, and the model demonstrated stable convergence. Additionally, we would like to point out that for DNT, we observed that it typically does not fail even when using larger learning rates, which further illustrates its strong stability.

---

> > ### Author Response · Authors · 2025-11-25
> > **Response to Reviewer epVN**
> >
> > We sincerely thank the reviewer for the time and effort in reviewing our paper. We have largely strengthened the paper according to your comments. We hope our clarifications and additional experiments address your concerns. Please let us know if any questions remain, and thank you again for your time and effort.
> >
> > Best regards
> >
> > Authors

---

### Official Review · Reviewer_rT6F · 2025-10-30

**Soundness:** 3
**Presentation:** 3
**Contribution:** 3
**Rating:** 4
**Confidence:** 3

**Summary:**

The paper proposes Deeply Normalized Transformer (DNT), a Transformer architecture designed to be effectively trained with momentum SGD (mSGDW) instead of adaptive optimizers like AdamW. The authors identify that Transformers exhibit heavy-tailed gradient distributions, which make SGD-based optimizers unstable. DNT addresses this by inserting or repositioning normalization layers (InputNorm, PreNorm, MidNorm, and QKNorm) to modulate the Jacobian of each block, ensuring more concentrated gradient distributions and reducing training instability.

**Strengths:**

1. Theoretical justification on how each normalization position affects the Jacobian and stabilizes gradient magnitudes.
2. Empirical results showing that DNT trained with mSGDW performs comparably to standard Transformers trained with AdamW, both on ImageNet (ViT) and OpenWebText (GPT2).

**Weaknesses:**

1. Theoretical assumptions are idealized: The high-dimensional isotropy and orthogonality assumptions may not hold exactly for real Transformer activations.
2. Similar ideas appear in nGPT, StableTransformer, and Lipsformer (which are cited), but the novelty claim is modest—it’s mostly a systematic integration and justification rather than a new normalization method.
3. The comparison is primarily between mSGD and AdamW. It would be more compelling to see how it performs against other optimizers like Sophia or Lion.
4. While comparing mSGDW and AdamW, the paper does not clarify if both optimizers were optimally tuned (learning rates, weight decay, warmup). The authors state "we did not tune the learning rate too much". However, the hyperparameter tables (e.g., Table 2, 4) show that the settings for mSGDW and AdamW are vastly different. For instance, L-DNT-Small uses LR=1.0 for mSGDW versus LR=6e-4 for AdamW , and V-DNT-Large uses LR=0.5 for mSGDW versus LR=1e-3 for AdamW.
5. In addition, There's also no evidence that DNT maintains benefits under fine-tuning, transfer, or longer training schedules.

**Questions:**

1. Could you include ablations isolating the impact of each normalization (InputNorm, PreNorm, MidNorm, QKNorm) individually on gradient statistics and performance?
2. How much GPU memory and wall-clock time are saved when training with mSGDW compared to AdamW?
3. How does DNT differ conceptually from “Transformers without normalization” (Zhu et al., 2025)? Could these approaches be unified?

---

> ### Author Response · Authors · 2025-11-22
> **Author responses to Reviewer rT6F (Part 1)**
>
> We would like to thank the reviewer for the constructive suggestions. To address your concerns, we have added more experiments, clarified our key contributions, and revised the paper according to your suggestions. We really hope you will be satisfied with our revisions.
>
> **Q1: Reviewer's comments on theoretical assumptions**
>
> > Theoretical assumptions are idealized: The high-dimensional isotropy and orthogonality assumptions may not hold exactly for real Transformer activations.
>
> **Our response:** Thank you very much for your comment. Let me explain our thought process: Through experiments, we observed the long-tail distribution issue in gradients during optimization. We concluded that effectively controlling the Jacobian matrix is key to addressing this long-tail problem. We then analyzed various normalization methods applied at different stages—specifically QKNorm, PreNorm, and MidNorm—by examining their Jacobian matrices and discussing their respective merits.
>
> We recognized the importance of ensuring that the norm of $\boldsymbol{x}$ does not become too large at every layer. It requires that the input to the first block is also properly normalized, leading us to propose InputNorm. For InputNorm, we incorporated the assumption of high-dimensional orthogonality.
>
> We believe high-dimensional probability serves as a powerful tool for interpreting Transformer behavior. The concentration inequality and the high-dimensional orthogonality in high-dimensional probability are relatively fundamental assumptions, and we consider them to be broadly satisfied under normal conditions. These assumptions are likely to break down only when the network encounters stability issues. That is why we should carefully consider how to put normalization layer to avoid this stability as much as possible.
>
> **Q2: On comparison with nGPT, StableTransformer, and Lipsformer**
>
> > Similar ideas appear in nGPT, StableTransformer, and Lipsformer (which are cited), but the novelty claim is modest—it’s mostly a systematic integration and justification rather than a new normalization method.
>
> __Our response:__ We want to clarify that, in this paper, we analyze the effects of normalization in different positions from a theoretical perspective. Specifically, we explain:
>
> - Why we need QKNorm for self-attention? QKNorm can relieve or even remove the influence of the magnitude of $\boldsymbol{W}_q$ and $\boldsymbol{W}_k$ on the Jacobian matrix of self-attention, and thus reduce the risk of model crash.
>
> - Why we need PreNorm for self-attention? Because PreNorm can constrain the norm of each column in activations $\boldsymbol{X}$ in each timestep, and thus amend the Jacobian matrix of self-attention to not be significantly affected by the magnitude of $\boldsymbol{X}$.
>
> - Why we need MidNorm? MidNorm will amend the Jacobian matrix of each sub-block (i.e., the sub-block with self-attention and the sub-block with FFN) in our DNT to not be affected by the magnitude of $\boldsymbol{W}_1$, $\boldsymbol{W}_2$, $\boldsymbol{W}_o$,$\boldsymbol{W}_v$.
>
>
> We would like to note that nGPT, StableTransformer, and LipsFormer do not explain where should we put a normalization or not?
>
> **Q3: On the empirical novelty**
>
> > The comparison is primarily between mSGD and AdamW. It would be more compelling to see how it performs against other optimizers like Sophia or Lion.
>
> **Our response:** Thank you for your suggestion. Since we have played Muon a while, we have conducted experiments using the Muon optimizer.
>
> We experimented with using the Muon optimizer to compare GPT-2 and DNT. Given an update $\boldsymbol{G}$, performing SVD decomposition, we have $\boldsymbol{G} = \boldsymbol{U} \boldsymbol{\Sigma} \boldsymbol{V}^{\top}$. As we know, Muon uses a Newton-Schulz method to approximate the $\boldsymbol{U} \boldsymbol{V}^{\top}$ matrix. In the current Muon implementation, for matrices, we use Muon to approximate the matrices, whereas for vectors, we directly employ the optimizer AdamW.}
>
> In our experiments with Muon, we used a learning rate of $10^{-4}$ for matrices and $6\times 10^{-4}$ for vectors. We conducted four sets of experiments: GPT-2 with AdamW, DNT with AdamW, GPT-2 with Muon, and DNT with Muon. Same as previous experiments, we use 2000 steps warmup for all four experiments.
>
> Based on experimental observations, we note the following two points:
> -1) When using the learning rates ($10^{-4}$, $6\times 10^{-4}$), GPT-2 begins to show an increase in validation loss after 50K training steps, indicating training instability. For DNT, we observe no instability issues, and the training curve exhibits almost no fluctuations. Additionally, when using Muon, DNT demonstrates significantly better convergence speed compared to DNT with AdamW.
> -2) When both DNT and GPT-2 use AdamW, DNT shows slightly better performance than GPT-2.

---

> ### Author Response · Authors · 2025-11-22
> **Author responses to Reviewer rT6F (Part 2)**
>
> **Q4: On mSGD and AdamW**
>
> > While comparing mSGDW and AdamW, the paper does not clarify if both optimizers were optimally tuned (learning rates, weight decay, warmup). The authors state "we did not tune the learning rate too much". However, the hyperparameter tables (e.g., Table 2, 4) show that the settings for mSGDW and AdamW are vastly different. For instance, L-DNT-Small uses LR=1.0 for mSGDW versus LR=6e-4 for AdamW , and V-DNT-Large uses LR=0.5 for mSGDW versus LR=1e-3 for AdamW.
>
> **Our response:** Thank you very much for your comment. We clarify the selection of parameters in Appendix C. For AdamW in ViT, we follow the standard Timm settings. For AdamW in GPT, we followed the settings in GPT3 and Sophia, we slightly tune AdamW around the suggested values and reported a better results. For our DNT, it is expensive to search the parameter in a large scale. For AdamW in V-DNT, we use the similar parameters as in ViT and GPT, but since our DNT can tolerent a larger learning rate, we use a larger learning rate, but we do not tune it to the best. You can see that, for L-DNT-Small, L-DNT-Large, and L-DNT-XL, we use 6e-4 for all three models. Be honest, it is hard for us to tune the parameter in fine scale.
>
> **Q5: On the transfer ability**
>
> > In addition, There's also no evidence that DNT maintains benefits under fine-tuning, transfer, or longer training schedules.
>
> **Our response:** We can see that some recent papers [1,2] primarily evaluate their models on GPT training tasks, with almost no assessment of fine-tuning and transfer capabilities. Therefore, we did not conduct related experiments in our work.
>
> Our model was trained for 200K steps. We note that many previous papers trained their models for only 100K steps or even fewer. We have made our best efforts to ensure very fair comparisons and analysis.
>
> [1] Wen, Kaiyue, David Hall, Tengyu Ma, and Percy Liang. "Fantastic pretraining optimizers and where to find them." *arXiv preprint arXiv:2509.02046* (2025).
>
> [2] Jordan, Keller, Yuchen Jin, Vlado Boza, You Jiacheng, Franz Cecista, Laker Newhouse, and Jeremy Bernstein. "Muon: An optimizer for hidden layers in neural networks, 2024." URL https://kellerjordan.github.io/posts/muon/.
>
> **Q6: On ablation study**
>
> > 1. Could you include ablations isolating the impact of each normalization (InputNorm, PreNorm, MidNorm, QKNorm) individually on gradient statistics and performance?
>
> **Our response:** Thank you very much for your suggestion. In the revised paper, we have added new experiments for ablation studies.
>
> In these experiments, we started with a complete DNT model that includes one InputNorm, two PreNorm layers, two MidNorm layers, and one QKNorm. We removed InputNorm, PreNorm (both layers), MidNorm (both layers), and QKNorm individually. We trained each ablated model with a relatively large learning rate (Adam optimizer with learning rate 0.3, without warmup) to observe when the training would collapse.
>
> We observed that:
>
> - When only removing QKNorm, the model collapsed after just a few training steps
> - When only removing PreNorm, the model began to collapse around 1000 steps
> - When only removing MidNorm, the model exhibited significant oscillations between 10K and 30K steps
> - When only removing InputNorm, the model was able to converge normally even with the 0.3 learning rate
>
> Therefore, from the perspective of training stability, we conclude that the importance of these normalization layers can be ranked in the following order: QKNorm > PreNorm $\approx$ MidNorm > InputNorm.
>
> Meanwhile, we also had an ablation study that started from the GPT baseline and then sequencially added each norm.
>
> **Q7: On the memory save**
>
> > 2. How much GPU memory and wall-clock time are saved when training with mSGDW compared to AdamW?
>
> **Our response:** We have trained 1.4B DNT model using both SGDW and AdamW. Theoretically, we can calculate that the memory taken by AdamW (only the optimizer part) is 11.5GB, and the memory costed by mSGDW (only the optimizer part) is 5.7GB. In the experiment, we obtained DNT+AdamW (model plus optimizer) costs 67GB, and DNT+mSGDW (model plus optimizer) costs 61GB. Using mSGDW instead of AdamW on 1.4B model can save around 6GB memory. Since the weight update is the computational bottleneck, our work can bring in wall-block time improvement.

---

> ### Author Response · Authors · 2025-11-22
> **Author responses to Reviewer rT6F (Part 3)**
>
> **Q8: On Transformers without normalization**
>
> > 3. How does DNT differ conceptually from “Transformers without normalization” (Zhu et al., 2025)? Could these approaches be unified?
>
> **Our response:** Actually, DyT from "Transformers without Normalization" can be unified with standard normalization methods. The essential purpose of normalization is to control the norm of $\boldsymbol{x}$ from becoming too large (such $10^5$), as excessively large norms of $\boldsymbol{x}$ can lead to attention collapse. First, let us explain why DyT can also be viewed as a form of normalization. DyT is defined as follows:
>
> $$
> \text{DyT}(x) = \gamma * \tanh(\alpha x) + \beta
> $$
>
> where $\gamma$ is initialized to 1, $\beta$ is initialized to 0, and $\text{DyT}(\cdot)$ is an element-wise operation applied to each dimension of the vector $\boldsymbol{x}$. If we reference RMSNorm, DyT can be written as: $\text{DyT}(x) = \gamma * \tanh(\alpha x)$.
>
> Let $\boldsymbol{y} = \text{DyT}(\boldsymbol{x})$, and let the dimension of $\boldsymbol{x}$ be $d$. We know that $\tanh(\alpha x) < 1$. If all $\gamma = 1$, then we have $\|\boldsymbol{y}\|_2 < \sqrt{d}$. This means that after passing through the DyT layer, regardless of the original norm of $\boldsymbol{x}$, the final output satisfies $\|\boldsymbol{y}\|_2 < \sqrt{d}$. Since $\boldsymbol{y}$ is fed into the self-attention module, its norm is effectively controlled, thereby preventing collapse in the self-attention mechanism.
>
> During actual operation, $\gamma$ is a learned parameter and does not remain constant at 1. Suppose we denote the maximum absolute value among the d-dimensional $\boldsymbol{\gamma}$ as $c$, then we have $\|\boldsymbol{y}\|_2 < c\sqrt{d}$. As long as the value of $c$ remains relatively bounded—for instance, less than 50—self-attention generally does not collapse. We have observed that when a network collapses during training, $\gamma$ tends to increase rapidly. However, by employing effective strategies such as learning rate warmup, careful placement of normalization layers, weight decay, etc., we can ensure stable and effective training.
>
> In summary, DyT also serves to control the activation $\boldsymbol{y}$ within a manageable range.
>
> Regarding the second question of why self-attention collapses when the norm of $\boldsymbol{x}$ or $\boldsymbol{X}$ is too large:
>
> Let us define
>
> $\boldsymbol{P} = \boldsymbol{X}^{\top} {\boldsymbol{W}_q}^{\top} {\boldsymbol{W}_k} {\boldsymbol{X}}$, $\boldsymbol{A} = \operatorname{softmax}(\frac{\boldsymbol{P}}{\sqrt{d_q}})$.
>
> If all $\boldsymbol{x}$ in $\boldsymbol{X}$ are large in norm, a phenomenon occurs where the attention matrix $\boldsymbol{A}$ becomes very sparse, and may even reduce to having only one entry of 1 per row, with all other entries being 0.
>
> We have $\boldsymbol{Y} = \boldsymbol{W}_v \boldsymbol{X} \boldsymbol{A}$. By vectorization of $\boldsymbol{Y} = \boldsymbol{W}_v \boldsymbol{X} \boldsymbol{A}$, we have
>
> $\partial {\text{vec}(\boldsymbol{Y})} = (\boldsymbol{A}^\top \otimes \boldsymbol{W}_v) \partial \text{vec}(\boldsymbol{X}) + (\boldsymbol{I}_n \otimes \boldsymbol{W}_v\boldsymbol{X}) \partial \text{vec}(\boldsymbol{A}).$
>
> According to matrix calculus, the Jacobian matrix is:
> $\frac{\partial \text{vec}(\boldsymbol{Y})}{\partial \text{vec}(\boldsymbol{X})} = (\boldsymbol{A}^\top \otimes \boldsymbol{W}_v) + (\boldsymbol{I}_n \otimes \boldsymbol{W}_v {\boldsymbol{X}}) \frac{\boldsymbol{J}}{\sqrt{d_q}} \left(
> ({\boldsymbol{X}^{\top}}{\boldsymbol{W}_k}^{\top}{\boldsymbol{W}_q} \otimes \boldsymbol{I}_n)\boldsymbol{C} + (\boldsymbol{I}_n \otimes {\boldsymbol{X}^{\top}}{\boldsymbol{W}_q}^{\top}{\boldsymbol{W}_k})
>  \right).$
>
> Please check the paper for the derivation. For simplicity, we denote:
>
> $$\boldsymbol{J} = \text{blockdiag}(\text{diag}(A_{:,1}) -A_{:,1} A_{:,1}^\top, \dots, \text{diag}(A_{:,n}) - A_{:,n} A_{:,n}^\top).$$
>
> where $n$ is the sequence length.
>
> If the attention matrix $\boldsymbol{A}$ contains only one 1 in each row with all other entries being 0, then we have $\text{diag}(A_{:,1}) - A_{:,1} A_{:,1}^\top \to \boldsymbol{0}$, then we know that this $\boldsymbol{J}$ matrix becomes entirely zero. Since the gradients for $\boldsymbol{W}_q$ and $\boldsymbol{W}_k$ are both propagated through the matrix $\boldsymbol{J}$, and $\boldsymbol{J}$ is zero, this means the gradients for $\boldsymbol{W}_q$ and $\boldsymbol{W}_k$ become zero. Consequently, the attention mechanism cannot update and collapses.
>
> In conclusion, DyT can be seen as a normalization strategy.

---

> > ### Author Response · Authors · 2025-11-25
> > **Response to Reviewer rT6F**
> >
> > We sincerely thank the reviewer for the time and effort in reviewing our paper. We have largely strengthened the paper according to your comments. We hope our clarifications and additional experiments address your concerns. Please let us know if any questions remain, and thank you again for your time and effort.
> >
> > Best regards
> >
> > Authors

---

### Official Review · Reviewer_QNDm · 2025-11-01

**Soundness:** 3
**Presentation:** 3
**Contribution:** 3
**Rating:** 6
**Confidence:** 4

**Summary:**

This paper proposes the Deeply Normalized Transformer (DNT), a Transformer variant designed to address the heavy-tailed gradient problem that hinders the performance of vanilla momentum SGD (mSGDW) in training Transformers. The authors provide theoretical analysis connecting the heavy-tail issue to the Jacobian matrices of different Transformer components and argue that strategically introducing normalization at specific positions (InputNorm, PreNorm, MidNorm, and QKNorm) can stabilize gradients and mitigate this issue. DNT is evaluated on both Vision Transformers (ViT) and GPT architectures, showing that it can be trained effectively with mSGDW to reach performance comparable to AdamW. The paper includes gradient distribution visualizations and empirical results on ImageNet and OpenWebText benchmarks.

**Strengths:**

- The paper tackles a practical and important problem—reducing dependency on adaptive optimizers like Adam—by improving Transformer architectures to work with simpler optimizers.

- The theoretical analysis provides a clear connection between normalization placement and Jacobian conditioning, offering intuition for the design of DNT. They also did a comprehensive analysis of different normalization techniques.

- Experimental results across both vision and language models demonstrate that DNT narrows the performance gap between mSGDW and AdamW, suggesting potential for simpler and more efficient training pipelines.

**Weaknesses:**

- The experiments, while promising, are limited in scale (e.g., GPT2-Small/Large, ViT-Large) and lack validation on larger models or diverse datasets to confirm robustness.
- The empirical novelty is modest, as the approach primarily reorganizes existing normalization techniques rather than introducing new mechanisms.
- The paper does not include detailed ablation studies to isolate the contribution of each normalization type, which would strengthen the empirical validation.

**Questions:**

- How does DNT scale when applied to very large LLMs (e.g., tens of billions of parameters)? Are there architectural or stability issues?
- Could the authors clarify whether the performance parity with AdamW holds when hyperparameters (e.g., learning rates, momentum) are more extensively tuned for mSGDW?
- How does DNT interact with modern optimizers such as Muon—does normalization reduce or amplify their benefits?
- What are the computational overheads introduced by the additional normalization layers in large-scale training?
- It appears that mSGDW with DNT generally performs worse than AdamW during the early stages of training—could the authors provide an explanation for this behavior?

Typo:
- line 158: any a forward layer

---

> ### Author Response · Authors · 2025-11-22
> **Author responses to Reviewer QNDm (Part 1)**
>
> We would like to express our sincere thanks to the reviewer for appreciating the contributions of this paper. *In this paper, we aim to deepen the theoretical understanding of Transformer models.*
>
> We have revised the paper according to your suggestions. And we also highlight the large-scale experiment in the main body. We also conduct more ablation studies to isolate the contribution of each normalization type.
>
> Hope our revisions and responses resolve your concerns.
>
> **Q1: Reviewer's comments on the scale of experiments**
>
> > The experiments, while promising, are limited in scale (e.g., GPT2-Small/Large, ViT-Large) and lack validation on larger models or diverse datasets to confirm robustness.
>
> **Our response:** Thank you very much for your suggestion. In the revised paper, we emphasized our large-scale experiments in Appendix E. We conducted experiments using a 1.4B language model, which we refer to as L-DNT-1.4B. Based on the experimental results, L-DNT-1.4B demonstrates better performance compared to its counterpart, GPT-2.
>
> Regarding the model size and dataset used in the experiments, I would like to note that in a recent and well-regarded optimization evaluation study [1], the model sizes evaluated ranged from 130M to 1.2B, and they evaluated results only on language modeling task. ImageNet and OpenWebText are two large-scale and representative datasets. In fact, training an L-DNT-1.4B model requires hundreds of GPU days. We hope the reviewers can take this into consideration.
>
> [1] Wen, Kaiyue, David Hall, Tengyu Ma, and Percy Liang. "Fantastic pretraining optimizers and where to find them." *arXiv preprint arXiv:2509.02046* (2025).
>
> __Q2: On the empirical novelty__
>
> > The empirical novelty is modest, as the approach primarily reorganizes existing normalization techniques rather than introducing new mechanisms.
>
> __Our response:__ Thank you very much for your comment. We believe that understanding something that already exists can sometimes be more important than inventing something new. In this paper, we aim to deepen the theoretical understanding of Transformer models, particularly the normalization components. To this end, we introduce a Deeply Normalized Transformer (DNT), which is meticulously engineered to address the issue of heavy-tailed gradients. In DNT, we strategically integrate normalization techniques at appropriate positions within the Transformers to effectively modulate the Jacobian matrices of each layer, balance the influence of weights, activations, and their interactions, and thereby concentrate the distributions of gradients. We also provide theoretical justifications for the normalization techniques employed in our DNT. *The core contribution of this paper lies in offering theoretical insights and design references for incorporating normalization at various positions in neural networks.*
>
> **Q3: On ablation study**
>
> > The paper does not include detailed ablation studies to isolate the contribution of each normalization type, which would strengthen the empirical validation.
>
> __Our response:__ Thank you very much for your suggestion. In the revised paper, we added more ablation studies in Appendix F. In these experiments, we started with a complete L-DNT model using AdamW that includes one InputNorm, two PreNorm layers, two MidNorm layers, and one QKNorm. We removed InputNorm, PreNorm (both layers), MidNorm (both layers), and QKNorm individually. We trained each ablated model with a relatively large learning rate (Adam optimizer with learning rate 0.3, without warmup) to observe when the training would collapse.
>
> We observed that:
>
> - When only removing QKNorm, the model collapsed after just a few training steps;
> - When only removing PreNorm, the model began to collapse around 4000 steps;
> - When only removing MidNorm, the model exhibited significant oscillations between 15K and 25K steps;
> - When only removing InputNorm, the model was able to converge normally even with the 0.3 learning rate.
>
> Therefore, from the perspective of training stability, we conclude that the importance of these normalization layers can be ranked in the following order: QKNorm > PreNorm $\approx$ MidNorm > InputNorm.
>
> Meanwhile, we also had an ablation study that started from the GPT baseline and then sequencially added each norm in the main body part.
>
> __Q4: Stability Issues on larger model__
>
> > How does DNT scale when applied to very large LLMs (e.g., tens of billions of parameters)? Are there architectural or stability issues?
>
> __Our response:__ Thank you for your comment. We ran experiments on a 7B model for 10K steps, and the model demonstrated stable convergence. Additionally, I would like to point out that for DNT, we observed that it typically does not fail even when using a very large learning rate (0.1 for AdamW), which further illustrates its strong stability.

---

> ### Author Response · Authors · 2025-11-22
> **Author responses to Reviewer QNDm (Part 2)**
>
> Q5: Parity of AdamW and mSGDW__
>
> > Could the authors clarify whether the performance parity with AdamW holds when hyperparameters (e.g., learning rates, momentum) are more extensively tuned for mSGDW?
>
> __Our response:__ Based on our experiments, we have observed that the performance gap between AdamW and mSGDW, in my opinion, still persists—even though DNT can help narrow this gap.
>
> __Q6: On DNT using Muon__
>
> > How does DNT interact with modern optimizers such as Muon—does normalization reduce or amplify their benefits?
>
> __Our response:__ Thank you for your suggestion. Following your advice, we have conducted experiments using the Muon optimizer. In experiments, we use the optimizer Muon to compare GPT-2 and DNT. Given an update $\boldsymbol{G}$, performing SVD decomposition, we have $\boldsymbol{G} = \boldsymbol{U} \boldsymbol{\Sigma} \boldsymbol{V}^{\top}$. As we know, Muon uses a Newton-Schulz method to approximate the $\boldsymbol{U} \boldsymbol{V}^{\top}$ matrix. In the current Muon implementation: for matrices, we use Muon to approximate the matrices; whereas for vectors, we directly employ the optimizer AdamW.}
>
> In our experiments with Muon, we used a learning rate of $10^{-4}$ for matrices and $6\times 10^{-4}$ for vectors. We conducted four sets of experiments: GPT-2 with AdamW, DNT with AdamW, GPT-2 with Muon, and DNT with Muon. Same as previous experiments, we use 2000 steps warmup for all four experiments.
>
> Based on experimental observations, we note the following two points:
> -1) When using the learning rates ($10^{-4}$, $6\times 10^{-4}$), GPT-2 begins to show an increase in validation loss after 50K training steps, indicating training instability. For DNT, we observe no instability issues, and the training curve exhibits almost no fluctuations. Additionally, when using Muon, DNT demonstrates significantly better convergence speed compared to DNT with AdamW.
> -2) When both DNT and GPT-2 use AdamW, DNT shows slightly better performance than GPT-2.
>
> __Q7: On computational overhead__
>
> > What are the computational overheads introduced by the additional normalization layers in large-scale training?
>
> __Our response:__ In our DNT, InputNorm is used only once, so it has very slight impact on computational speed. In our DNT architecture, we employ 5 or 6 normalization layers in total. Compared to the standard GPT-2 model, we use 3 additional normalization layers, resulting in a computational overhead that is 6% higher than GPT-2. When compared to a GPT-2 model that uses QKNorm (which employs 4 normalization layers), our computational overhead is only 2% higher. Overall, the additional computational overhead is relatively small.
>
> __Q8: On the early stage of training of DNT using mSGDW and AdamW__
>
> > It appears that mSGDW with DNT generally performs worse than AdamW during the early stages of training—could the authors provide an explanation for this behavior?
>
> __Our response:__ We assume that $\boldsymbol{U}_t$ is the update at time step $t$. In **mSGDW**, $\boldsymbol{U}_t$ is obtained directly from the gradient with a momentum, while in AdamW, $\boldsymbol{U}_t$ is derived by dividing the first moment by the square root of the second moment.
>
> It can be observed that in mSGDW, the singular values of $\boldsymbol{U}_t$ are unbounded, and their distribution tends to concentrate along certain directions — that is, some values are very large while others are very small. In contrast, in AdamW, the values of $\boldsymbol{U}_t$ are generally bounded, and the distribution of its singular values is more uniform compared to mSGDW, meaning it retains meaningful values across more directions.
>
> In other words, compared to mSGDW, AdamW performs gradient updates across more directions, whereas mSGDW tends to focus heavily on only a few directions. Roughly speaking, the update of mSGDW is low-rank, while that of AdamW has a higher rank.
>
> This is the reason why, in the early stages, mSGDW with DNT generally performs worse than AdamW. To some extent, our proposed DNT helps alleviate this problem by mitigating the issue of heavy-tail gradients.
>
> __Q9: On typos__
>
> > line 158: any a forward layer
>
> __Our response:__ Thank you for pointing out this typo, we have corrected it.

---

> > ### Author Response · Authors · 2025-11-25
> > **Response to Reviewer QNDm**
> >
> > We sincerely thank the reviewer for the time and effort in reviewing our paper. We have largely strengthened the paper according to your comments. We hope our clarifications and additional experiments address your concerns. Please let us know if any questions remain, and thank you again for your time and effort.
> >
> > Best regards
> >
> > Authors

---

### Author Response · Authors · 2025-11-22
**Summary of our revisions**

We would like to thank all reviewers for their time and effort in reviewing our paper.

To address the reviewers' questions or concerns, we have made the following revisions:

- We added more ablation studies to isolate the contribution of each normalization type as suggested by reviewers rT6F and gReu.

- We added experiments using Muon as our optimizer as suggested by reviewers QNDm, rT6F, and epVN.

- We emphasized our L-DNT-1.4B large-scale language model experiment on OpenWebText.

- We revised the paper, updated the figures, clarified some notations when we first mentioned them according to the suggestions from the reviewer gReu.

The revised contents are highlighted in blue in the revised paper. **We kindly invite all reviewers to take a look at our new experiments.**

We sincerely thank all reviewers again for their valuable suggestions, which have greatly helped strengthen our paper.

If you have any further questions, we would be happy to discuss them.

---

### Meta-Review · Area_Chair_931f · 2026-01-06

**Summary:**

This paper proposes Deeply Normalized Transformers (DNT), which place normalization at Input, Pre, Mid, and QK locations to tame heavy-tailed gradients and make mSGDW viable for training Transformers. The analysis links each normalization placement to improved Jacobian conditioning. Experiments on ViT/ImageNet and GPT/OpenWebText show that DNT narrows the gap between mSGDW and AdamW. The rebuttal strengthens the work with per-normalization ablations, clearer notation, added Muon results showing stability and faster convergence with DNT, and a highlighted 1.4B-parameter experiment. The idea is practical and well motivated.

However, the empirical scope remains modest for the claims. Evidence at larger scales is limited, with no thorough transfer, fine-tuning, or broader benchmarks. Several stronger baselines are missing or lightly tuned (e.g., Sophia, Lion; stability architectures like nGPT, Stable Transformer, LipsFormer), and hyperparameter parity between mSGDW and AdamW is not fully established. The contribution is mainly a systematic integration and justification of known normalization ideas. The manuscript still shows minor clarity issues that need polishing.

**Reviewer Concerns:**

Addressed: Added ablations isolating InputNorm, PreNorm, MidNorm, QKNorm; clearer notation and references; Muon experiments indicating improved stability with DNT; compute/memory overhead estimates; explanation for early-stage mSGDW behavior.

Outstanding: Limited external validity beyond the reported setups; lack of strong matched baselines and tuning parity; no systematic scaling or transfer/fine-tuning evidence; novelty is largely architectural reorganization; remaining presentation polish.

**Reviewer Scores:**

The strongest reviewer likely stays at 6 given the added experiments.

Reviewers at 4 are likely to remain at 4 without broader baselines or larger-scale validation.

Personally, I was impressed by the responses to reviewers, but reviewers did not follow up with updates on their scores.

---

### Decision · Program_Chairs · 2026-01-26

Accept (Poster)